# Mesophotic and Bathyal Palaemonid Shrimp Diversity of the Red Sea, with the Establishment of Two New Genera and Two New Species [†]

Arthur Anker [1,2], Silvia Vimercati [2,3], Federica Barreca [2], Fabio Marchese [2], Giovanni Chimienti [2,4], Tullia I. Terraneo [2], Mattie Rodrigue [5], Ameer A. Eweida [6], Mohammed Qurban [7], Carlos M. Duarte [2,3], Vincent Pieribone [5] and Francesca Benzoni [2,3,*]

1. Instituto de Biologia, Campus do Capão do Leão, Universidade Federal de Pelotas (UFPEL), Prédio 5, Capão do Leão 96160-100, Rio Grande do Sul, Brazil; arthuranker7@gmail.com
2. Red Sea Research Center (RSRC), Division of Biological and Environmental Science and Engineering, King Abdullah University of Science and Technology (KAUST), Thuwal 23955-6900, Saudi Arabia; silvia.vimercati@kaust.edu.sa (S.V.); federica.barreca@kaust.edu.sa (F.B.); fabio.marchese@kaust.edu.sa (F.M.); giovanni.chimienti@uniba.it (G.C.); tulliaisotta.terraneo@kaust.edu.sa (T.I.T.); carlos.duarte@kaust.edu.sa (C.M.D.)
3. Marine Science Program, Division of Biological and Environmental Science and Engineering, King Abdullah University of Science and Technology (KAUST), Thuwal 23955-6900, Saudi Arabia
4. Consorzio Nazionale Interuniversitario per le Scienze del Mare (CoNISMa), 00196 Rome, Italy
5. OceanX, 37 West 39th St., New York, NY 10018, USA; mattie@oceanx.org (M.R.); vincent@oceanx.org (V.P.)
6. Center of Carbonate Research, Department of Marine Geosciences, Rosenstiel School of Marine and Atmospheric Science, University of Miami, Miami, FL 33149, USA; ameer.eweida@gmail.com
7. National Center for Wildlife, Riyadh 11575, Saudi Arabia; mqurban@mewa.gov.sa
* Correspondence: francesca.benzoni@kaust.edu.sa
† New Species: urn:lsid:zoobank.org:pub:6F151518-D680-40F8-B187-04B09FF70F7F.

**Abstract:** The diversity and evolution of the Red Sea invertebrates in mesophotic and deep-water benthic ecosystems remain largely unexplored. The Palaemonidae is a diversified family of caridean shrimps with numerous taxa in need of taxonomic revisions based on recent molecular analyses. The Red Sea mesophotic and bathyal palaemonid shrimps are largely unstudied. During recent expeditions off the Red Sea coast of Saudi Arabia, several palaemonid specimens were collected at a depth range of 88–494 m, spanning the mesophotic and bathyal zones. This material was examined morphologically and genetically to infer phylogenetic relationships among the Red Sea taxa and several other palaemonid genera. The concordant morphological and genetic data led to the description of two new genera and two new species. Moreover, one species was recorded in the Red Sea for the first time, with a new host record, whereas three further deep-water species, which do not occur in the Red Sea, were formally transferred to a different genus. As more exploration efforts are deployed, research on the diversity and evolutionary relationships among marine invertebrates from the Red Sea will further underline the uniqueness of its mesophotic and bathyal fauna.

**Keywords:** marine biodiversity; mesophotic benthos; crustaceans; symbiosis; phylogeny; deep-water shrimps; Palaemonidae; new genus; new species; new combination; new record

## 1. Introduction

The Red Sea is a young semi-enclosed ocean [1], in which shallow, mesophotic and deep environmental conditions are extreme, especially due to the high water temperature, salinity and, below a certain depth, oxygen levels [2]. It is therefore relatively unsurprising that the Red Sea hosts one of the highest levels of endemism known for shallow marine habitats worldwide that likely evolved in response to the unique mix of geological history and ecological gradients in the basin [2,3]. The same may be true for the deep-water habitats of the basin, although the presently available biological data are increasingly more limited

with depth [4]. Below 500 m depth, the Red Sea benthos appears to be characterised by high endemism and a significantly deeper bathymetry of numerous common species compared with the rest of the Indo-West Pacific [4,5]. The sampling efforts on the deeper component of the Red Sea fauna, i.e., below scuba-diving limits, are, however, extremely scanty at the scale of the basin and are limited to surveys that mostly predate the advent of the last two decades of technology in field and laboratory tools and analyses.

The shrimp family Palaemonidae, which is the largest caridean family approaching 1000 species, is known mainly due to the economic importance of several freshwater species [6]. The popularity of the Palaemonidae among scuba divers and underwater photographers is also worth mentioning. In addition, numerous marine palaemonid species live in association with an impressive array of other benthic invertebrates, which makes them excellent models for studying various aspects of the evolution of symbioses [7,8].

The mesophotic and deep-water (below 50 m depth) palaemonid shrimps of the Red Sea are overall poorly known. Bruce & Svoboda [9] summarised palaemonid records from the shallow parts of the Red Sea and reported several additional shallow-water species from the Gulf of Aqaba. Similarly, all palaemonid shrimps listed by Ďuriš [10] from Saudi Arabia are shallow-water species. Bruce [11,12] recorded *Periclimenes pholeter* Holthuis, 1973 from 1825 to 2148 m in the Red Sea, where the species appears to be confined to two extreme habitats, deep-sea brine pools and deep, near-shore, anchialine cracks [13,14]. This morphologically and ecologically very distinctive species will likely be removed from *Periclimenes* Costa, 1844 (*sensu stricto*); however, its phylogenetic affinities presently remain unclear. Bruce [15] described *Palaemonella meteorae* Bruce, 2008 from 519 to 544 m based on two damaged specimens, with all pereiopods missing. Since then, to our best knowledge, there have been no records of deep-water palaemonid shrimps from the Red Sea and/or the adjacent Gulf of Aden.

The present study deals with part of the palaemonid material collected by several research cruises onboard the R/V OceanXplorer off the coast of Saudi Arabia in 2020 and 2022. The main goal of these expeditions was to obtain the first overview of the country's marine mesophotic and bathyal diversity and resources. The herein reported palaemonid material is composed of: (1) two specimens of a species originally described as *Periclimenes foveolatus* Bruce, 1981; (2) several specimens of a new species of the recently proposed genus *Michaelimenes* Okuno 2017; (3) two specimens of a new species initially assigned to *Periclimenes* and morphologically most similar to *P. kallisto* Bruce, 2008 and *P. affinis* (Zehntner, 1894); and (4) a single specimen of *Apopontonia falcirostris* Bruce, 1976 [16–20]. A molecular phylogeny reconstruction based on two mitochondrial markers (COI and 16S) was performed by incorporating gene sequences obtained from the first three taxa into the *Periclimenes s. l.* dataset from the three most relevant molecular studies of the Palaemonidae [8,21,22].

## 2. Materials and Methods

### 2.1. Sampling

The material examined in this study was collected at multiple stations along the Saudi Arabian Red Sea coast Neom–OceanX Deep Blue Expedition in 2020, the Saudi Arabia National Center for Wildlife (NCW)–OceanX Red Sea Decade Expedition in 2022 and the OceanX Red Sea Relationships Cultivation Expedition in 2022, aboard M/V OceanXplorer. Deep benthos collection took place using Chimaera (specimen and station codes CHR), an Argus Mariner XL108 remotely operated vehicle (ROV), and Neptune (specimen and station codes NTN), a Triton 3300/3 submersible. The ROV equipment included a Kongsberg HiPaP 501 Ultra Short Base Line acoustic tracking system and a complex light-cameras apparatus with, among others, one DSPL Super wide-angle CCD camera for landscape view and one HDTV 1080p F/Z colour camera for detailed observations. Each submersible was equipped with a Sonardyne Ranger Pro 2 system and several light and camera systems including a Wide Angle Red DSMC2 Helium 8k Canon CN-E15.5–47 mm lens and a macro Red DSMC2 Helium 8k Nikon ED 70–180 mm F4.5-5.6D. Both the ROV and submersibles were also equipped with a CTD probe (RBR Maestro CTD and Sea-Bird SBE 19+, respectively), two parallel-aligned scaling lasers providing 100 mm scale and a

Schilling T4 hydraulic manipulator for sampling. The ROV and submersible dives were video-recorded, and frame grabs of the shrimp hosts were later extracted from the videos.

### 2.2. Morphological Analyses and Measurements

After sampling, some specimens were photographed alive using an Olympus Tough TG-6 waterproof camera and then preserved in 70% ethanol. Observations of the external morphology and drawings were made under two different stereomicroscopes equipped with a camera lucida. Postorbital (pocl) and total (cl) carapace lengths were measured in mm along the mid-dorsal line from the posterior orbital margin and rostral tip to the posterior margin of the carapace, respectively. For the endopod of the third maxilliped, the terms antepenultimate, penultimate and ultimate articles are used for ischio-merus, carpus and propodo-dactylus, respectively.

### 2.3. Material Registration and Remarks on Type Material

The examined material was deposited in the collections of the Florida Museum of Natural History, University of Florida, Gainesville, FL, USA (FLMNH UF), and Red Sea Research Center Reference Collection, King Abdullah University of Science and Technology, Thuwal, Saudi Arabia (RSRC). Unfortunately, some of the studied material, including the originally designated and drawn type specimens, was subsequently lost during transportation. In the case of the new species of *Michaelimenes*, two paratypes, including the only male specimen, were lost; however, the holotype and several other paratypes were not affected. In the case of the new species assigned to a new genus, both the holotype and the paratype were lost. Our justification for exceptionally describing a new species without extant type material is based on four facts: (1) the loss of the type material occurred after a thorough morphological study of each specimen, i.e., determination of sex, all required measurements, morphological observation and completion of numerous line-drawings; (2) the new species was morphologically clearly distinguishable from all other species described in the polyphyletic generic complex around *Periclimenes*; (3) the knowledge of the host of the new species will certainly help in its recollection in the future; and (4) one of the specimens was subsampled and sequenced, with genetic data (COI, 16S, 18S, H3) deposited in GenBank (see Supplementary Material Table S1).

### 2.4. DNA Extraction, Amplification and Sequence Analyses

Total DNA was extracted from pleopod or gill tissue using a DNeasy® Blood and Tissue Kit (Qiagen Inc., Hilden, Germany), following the manufacturer's protocol. The mitochondrial genes for cytochrome c oxidase subunit I (COI) and 16S RNA were amplified using the primer pairs LCO1490/HCO2198 [23] and 16Sar/16Sbr [24], respectively. The nuclear genes H3 and 18S rRNA were amplified using the primer pairs H3F/H3R [25] and 18Sa2.0/18S9r [26], respectively. The amplification was performed in a 15 μL volume of 1X Multiplex PCR Master Mix (Qiagen Inc., Hilden, Germany), 0.2 μM of each primer and <5 ng DNA. The thermal cycling profiles of Horka et al. [8] were followed. The amplification success was tested using a QIAxcel Advanced System (Qiagen Inc., Hilden, Germany). The amplified products were purified using Illustra Exostar (GE Healthcare, Buckinghamshire, UK), following the manufacturer's protocol, and directly sequenced in forward and reverse directions using an ABI 3730xl DNA analyser (Applied Biosystems, Foster City, CA, USA). Forward and reverse sequences were edited and assembled using BioEdit Sequence Alignment Editor 7.2.6 [27] and Geneious® v.10.1.3 (Biomatters Ltd., Auckland, New Zealand). *Stenopus hispidus* (Olivier, 1811) (Stenopodidae) was selected as an outgroup based on previously published molecular data [8]. Additional sequences from GenBank included in the analyses [21,28,29] are listed in Supplementary Material Table S1. The final alignment was performed using the E-INS0i strategy in MAFFT 7 [30] under default parameters. All sequences produced in this study were deposited in GenBank (https://www.ncbi.nml.nih.gov/genbank/ sequences deposited on 5 June 2023) with accession numbers listed in Supplementary Material Table S1. Bayesian inference (BI) and

maximum likelihood (ML) were used to infer the phylogeny, whereas the Akaike information criterion (AIC) with Partition Finder 2.1 [31] was used to determine the sequence evolution best-fit substitution model. The BI analysis was run using MrBayes 3.2.7a [32], using 10 million generations, saving a tree every 1000 generations and discarding the first 2500 trees as burn-in, based on parameter estimations and convergence examined using Tracer 1.7.1 [33]. The ML analysis was run using RaxML 8.2.12 [34] with default parameters and 1000 bootstrap interactions to verify the robustness of the internal branches of the tree. BI and ML analyses were run on the CIPRES server [35].

## 3. Results and Discussion

The herein-reported palaemonid material was collected at several localities spanning the north, central and south-central Red Sea off Saudi Arabia (Figure 1A), during five manned submersible dives and five ROV dives (Figure 1B). The shrimps were picked off blocks of substrate or larger invertebrate hosts (sponges or cnidarians) sampled between 88 m and 494 m depth, i.e., within a depth range encompassing mesophotic and upper bathyal conditions. The material contained four species belonging to four genera, including two new species and two new genera (see below). All genera and species are formally reported or described in the Systematics Section hereafter.

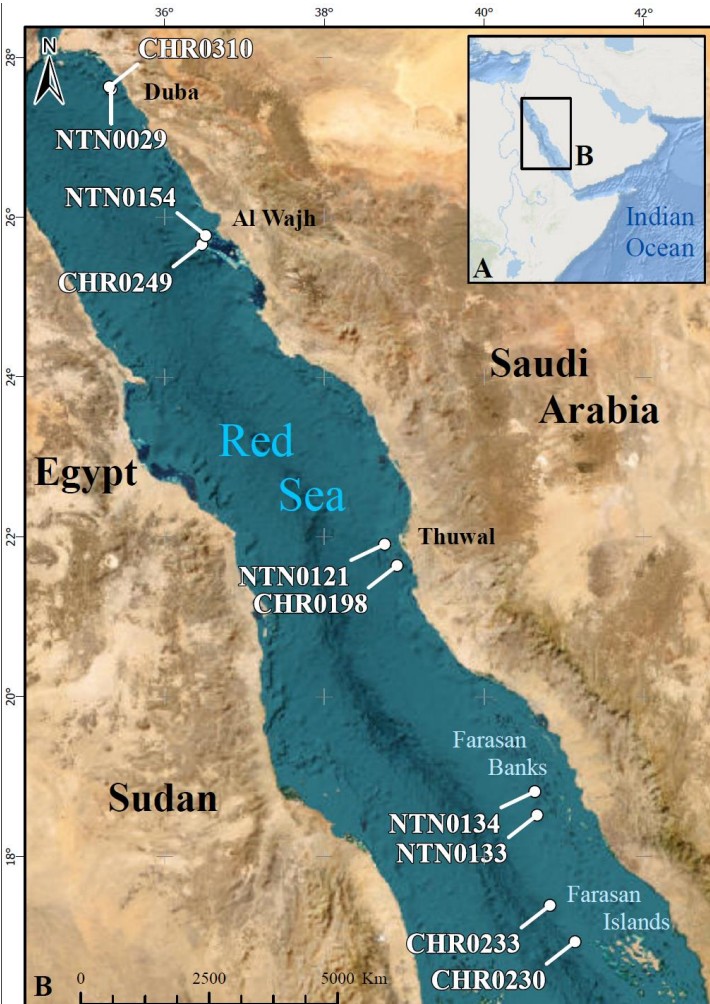

**Figure 1.** Map of the Red Sea (**A**) showing the sampling area of this study (**B**), where the sampling stations for material reported in this study are shown. Codes indicate the submersible (NTN) or remotely operated vehicle (CHR) dives from the Deep Blue Expedition (2020), Red Sea Decade Expedition (2022) and Red Sea Relationships Cultivation Expedition (2022). Details of each sampling site are provided in the taxonomic section. Basemap credits: Esri, GEBCO, DeLorme, NaturalVue and Earthstar Geographics.

*3.1. Phylogenetic Analyses*

In total, 22 sequences were obtained for COI (5), 16S (6), 18S (6) and H3 (5) markers (Supplementary Material Table S1). In addition, 119 sequences for COI (29), 16S (35), 18S (23) and H3 (32) markers were retrieved from GenBank [8,21,28,29]. The final four loci concatenated alignment (COI, 16S, 18S and H3) consisted of 2133 bp (658 for COI, 465 bp for 16S, 662 bp for 18S and 348 bp for H3), including 1305 conservative sites and 818 variable sites, of which 661 were parsimony informative and 157 were singletons.

The obtained phylogenetic tree (Figure 2) includes representatives of 12 previously described palaemonid genera, including several deep-water taxa, and two newly proposed genera (see below). The Red Sea material was retrieved in three distinct and well-supported clades (Figure 2, OCX-labelled specimens). In agreement with previous results (e.g., [8]), available sequences for representatives of several species currently ascribed to *Periclimenes* were retrieved in six different clades or lineages, thus confirming the polyphyletic status of this genus. The small eastern Atlantic clade containing the type species of *Periclimenes*, *P. amethysteus* (Risso, 1827) (Figure 2, highlighted in beige), represents *Periclimenes s. str.* (e.g., [8,36]).

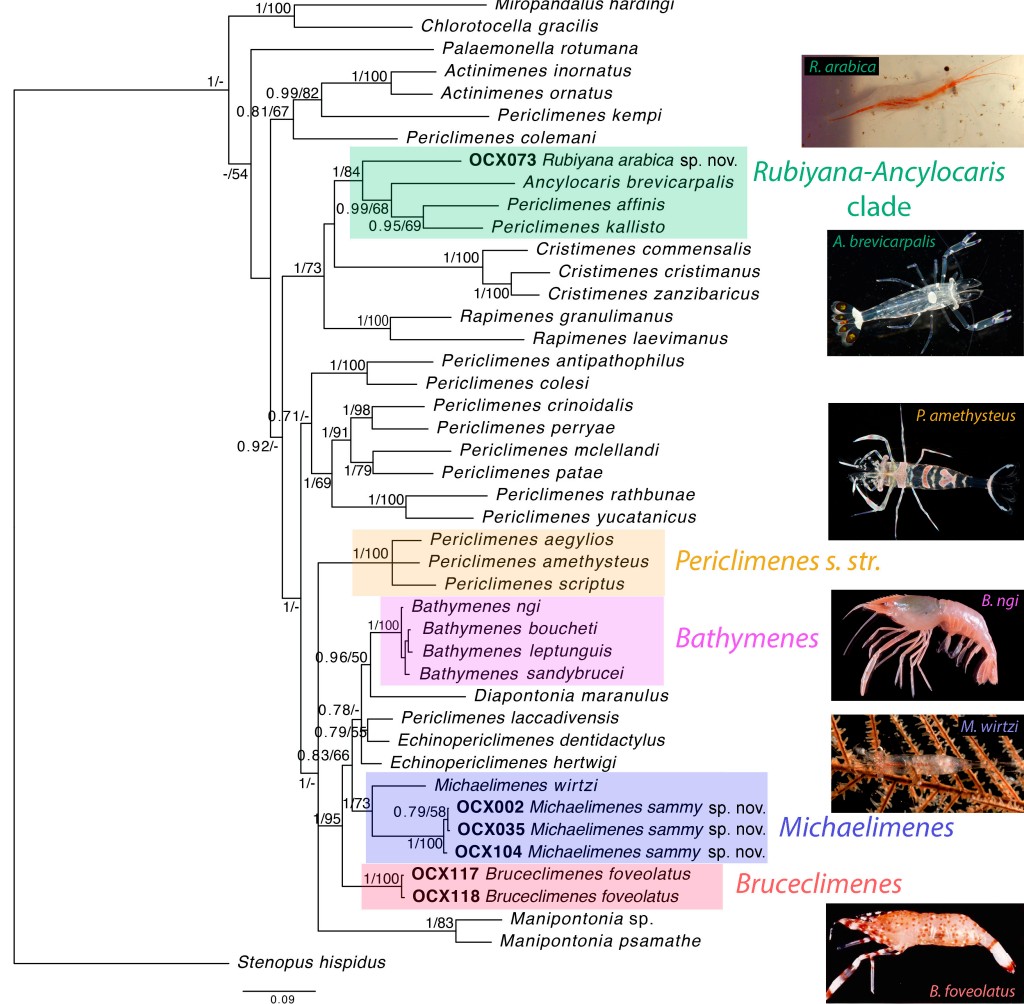

**Figure 2.** Phylogeny reconstruction of several clades within *Periclimenes* Costa, 1844 *sensu lato*, including the Red Sea material collected by the OceanX (OCX) expeditions, *Michaelimenes sammy* sp. nov., *Bruceclimenes foveolatus* (Bruce, 1981) comb. nov. and *Rubiyana arabica* gen. et sp. nov., and highlighting *Periclimenes s. str.* and *Bathymenes* Kou, Li & Bruce, 2016, discussed in more detail in the text, inferred from Bayesian inference (BI) analysis of the combined mitochondrial (COI, 16S) and nuclear (18S, H3) datasets. The Chlorotocellidae Komai, Chan & De Grave, 2019 family lineage is represented here by representatives of the genera *Miropandalus* Bruce, 1983 and *Chlorotocella* Balss, 1914 at the top of

the tree. Node values are posterior Bayesian probabilities and maximum likelihood (ML) bootstrap values. Posterior Bayesian probabilities below 0.7 and ML bootstrap values below 50% are indicated with a dash (-). *Stenopus hispidus* (Stenopodidae) was used as the outgroup. Insert photographs by F. Barreca (*R. arabica*), A. Anker (*A. brevicarpalis*, *P. amethysteus*), T.Y. Chan (*B. ngi*), S. Lobenstein (*M. wirtzi*) and P. Laboute (*B. foveolatus*).

Two specimens from the Red Sea (OCX117 and OCX118) initially identified as *Periclimenes foveolatus*, a deep-water species described by Bruce [18] from the Philippines and later reported from New Caledonia [37], were retrieved as a distinct genus-level lineage. This lineage is hereafter described as *Bruceclimenes* gen. nov. (authored by Anker), with the type species *Bruceclimenes foveolatus* (Bruce, 1981) comb. nov. The obtained tree topology, which must be considered preliminary due to the voluntary omission of several palaemonid genera that were non-essential for the present discussion but were included in the analysis of Horká et al. [8], indicates no close relationship between *Periclimenes s. str.* and *Bruceclimenes* (Figure 2). In fact, *Bruceclimenes* is part of a larger clade composed of several genera, which also contains *Michaelimenes*, represented by three specimens of a new species from the Red Sea (OCX002, OCX035 and OCX104) described below as *M. sammy* sp. nov. Furthermore, the molecular analysis provided good support for *M. sammy* being in a sister position to *Periclimenes wirtzi* d'Udekem d'Acoz, 1996, an eastern Atlantic species with many diagnostic features of *Michaelimenes*. Therefore, *P. wirtzi* is formally transferred to *Michaelimenes*, as *Michaelimenes wirtzi* (d'Udekem d'Acoz, 1996) comb. nov., and the diagnosis of the genus is emended. The specimen OCX073, which is one of two specimens preliminarily assigned to a new species of *Periclimenes*, was retrieved within a lineage comprising *Periclimenes affinis* (Zehntner, 1894), *P. kallisto* Bruce, 2008 and *Ancylocaris brevicarpalis* Schenkel, 1902 (Figure 2). This clade is even more distant from the clade representing *Periclimenes s. str.* (Figure 2; see also above). Therefore, a new genus, *Rubiyana* gen. nov. (authored by Anker), is established for the new species and the phylogenetic positions of the other members of this clade are discussed below. Finally, some taxonomic and biogeographic notes are provided for the minute and rarely reported, sponge-associated *Apopontonia falcirostris*.

*3.2. Systematics*

**Palaemonidae Rafinesque, 1815**

***Bruceclimenes* gen. nov.** (authored by Anker)
(urn:lsid:zoobank.org:act:898DB255-7A72-411B-A886-34B4AA5F4303)

*Diagnosis*: Medium-sized palaemonid shrimps (pocl 4.9–9.5 mm). Body subcylindrical, slightly depressed, smooth; lateral surface of carapace and pleon alveolate, sometimes less distinctly. Rostrum well developed, high, somewhat descending, exceeding antennular peduncle; dorsal margin convex, with 8–10 teeth, posterior-most tooth in postorbital position; ventral margin convex, with 3–6 teeth in distal half; rostral carina strong, continuing posteriorly as elevated mid-dorsal carina, lowering before reaching posterior margin of carapace. Inferior orbital angle strongly anteriorly produced, reaching beyond antennal tooth, blunt or subacute. Antennal and hepatic teeth well developed, acute; latter stouter and situated slightly below former, not reaching anterolateral margin of carapace. Epigastric and supraorbital teeth absent. Thoracic sternite 4 without median process. Pleonite 3 not carinate; posterior margin not posteriorly produced; pleura of pleonites 1–5 posteroventrally rounded; sixth pleonite not elongate. Telson with two pairs of minute, submarginal, dorsal spiniform setae in posterior half and three pairs of short spiniform setae on posterior margin, mesial ones with secondary setules; posterior margin with minute, central tooth between mesial spiniform setae. Antennule typical for family; stylocerite short, acute; distolateral tooth of first article of peduncle sharp; lateral flagellum biramous, with long fused basal part; secondary ramus well developed. Antenna typical for family; scaphocerite well developed, with broad blade and stout distolateral tooth falling short of anterior margin

of blade; carpocerite short. Eyes well developed, with large globular cornea. Mandible without palp; molar and incisor processes normal. Maxillule with deeply bilobed palp (endopod), ventral lobe with minute, hooked seta. Maxilla with reduced coxal endite and bilobed basial endite; palp (endopod) small; scaphognathite broad. Maxilliped 1 with endites fused; palp (endopod) small, entire; exopod long, with well-developed caridean lobe; epipod moderately large, subtriangular. Maxilliped 2 with normal endopod and exopod; epipod rounded, without podobranch. Maxilliped 3 with coxa bearing broad, semicircular, lateral plate; endopod robust; antepenultimate article clearly separated from basis, with thick setae on ventral margin; ultimate article with stiff setae; exopod well developed; single arthrobranch moderately developed. Pereiopod 1 slender; carpo-propodal brush present; chela not elongated, not inflated, with fingers as long as palm; cutting edges of fingers unarmed. Pereiopods 2 moderately stout, similar in shape, subequal in size; merus and carpus unarmed, minutely tuberculate; carpus cup-shaped; chela not particularly swollen; palm subcylindrical, moderately elongate, densely covered with minute tubercles; fingers much shorter than palm; cutting edges of fingers each armed with two large teeth proximally, without tooth–fossa snapping mechanism. Pereiopods 3–5 moderately slender; propodus with small spiniform setae on ventral margin; dactylus subtly biunguiculate, with minute secondary unguis on stout corpus and slender, curved, terminal unguis. Male pleopod 1 with endopod simple, expanded distally; lateral margin with plumose setae; mesial margin with robust plumose setae and short spinules; distal margin with two feeble, simple setae. Male pleopod 2 with appendix masculina not overreaching appendix interna, with three longitudinal rows of stiff, simple setae. Uropod normal; protopod with two bluntly ending lobes; distolateral spiniform seta very small; endopod narrower and shorter than exopod.

*Type species*: *Bruceclimenes foveolatus* (Bruce, 1981) comb. nov., originally described as *Periclimenes foveolatus* Bruce, 1981, by present designation and monotypy.

*Other species included*: None.

*Etymology*: The name of the new genus combines the last name of Dr. Alexander J. Bruce (1929–2022), an eminent carcinologist, who described hundreds of Indo-Pacific palaemonid shrimps (including *P. foveolatus*, its type species, see above), and the second part of the word *Periclimenes*, one of the oldest and largest palaemonid genera. Gender masculine.

*Hosts:* Deep-water sea anemones (Actiniaria).

*Bathymetric range*: 187–471 m.

*Distribution*: Indo-West Pacific from the Red Sea to the Philippines and New Caledonia.

*Remarks*: The stepwise revision of the problematic genus *Periclimenes* (e.g., [8,22,38]) resulted in the description of new genera and revalidation of several previously synonymised genera over the last two decades (e.g., [10,20,36,39–44]). The molecular analysis of selected, mainly symbiotic palaemonid taxa by Horká et al. [8] showed multiple host switches and adaptive convergences, which impeded the genus-level taxonomy within this group. *Periclimenes* was, not surprisingly, recovered as a complex, polyphyletic assemblage. Similarly, Frolová et al. [45] did not recover monophyletic *Cuapetes* Clark, 1919 and *Palaemonella* Dana, 1852, with several symbiotic taxa belonging to other genera, such as *Vir* Holthuis, 1952 and *Eupontonia* Bruce, 1971, embedded within these genera, whereas some free-living taxa were found in different parts of the tree. These two studies, as well as molecular analyses by Gan et al. [21] and Kou et al. [22,28,29], using different datasets, show that numerous taxonomic changes, including redefinitions of larger genera and descriptions of new genera, are expected in the Palaemonidae.

The morphological distinctiveness of the species originally described as *Periclimenes foveolatus* was recognised by Bruce [18] and is confirmed by our molecular analysis (Figure 2), justifying the establishment of a new genus for this strikingly coloured and ecologically unique species associated with deep-water sea anemones (Figure 3D). *Bruceclimenes* can be separated from *Periclimenes s. str.* and related genera by the combination of the following characters: (1) carapace with strong, elevated, mid-dorsal carina, extending almost to its posterior margin; (2) hepatic tooth not greatly enlarged and not reaching or

overreaching anterolateral margin of carapace; (3) lateral surface of carapace and pleon with a more or less pronounced "foveolation" (term used by Bruce [37] for a surface characterised by the presence of numerous small depressions or pits); (4) third pleonite not forming a hump, i.e., not carinate or posteriorly produced; (5) dorsal surface of telson with two pairs of minute spiniform setae; (6) mandible without palp; (7) cutting edges of first pereiopod fingers blade-like, unarmed; (8) surface of second pereiopods finely tuberculate; and (9) dactylus of third to fifth pereiopods subtly biunguiculate without further specialised structures. This combination of characters separates *Bruceclimenes* from all other genera allied to *Periclimenes*, including *Bathymenes* Kou, Li & Bruce, 2016 (see remarks below), *Echinopericlimenes* Marin & Chan, 2014, *Michaelimenes* Okuno, 2017, *Diapontonia* Bruce, 1986 and *Zoukaris* Anker & Corbari 2020, which are genera containing mainly deep-water species [20,36,44,46] as well as *Ancylocaris* Schenkel, 1902, a well-known associate of shallow-water sea anemones [10].

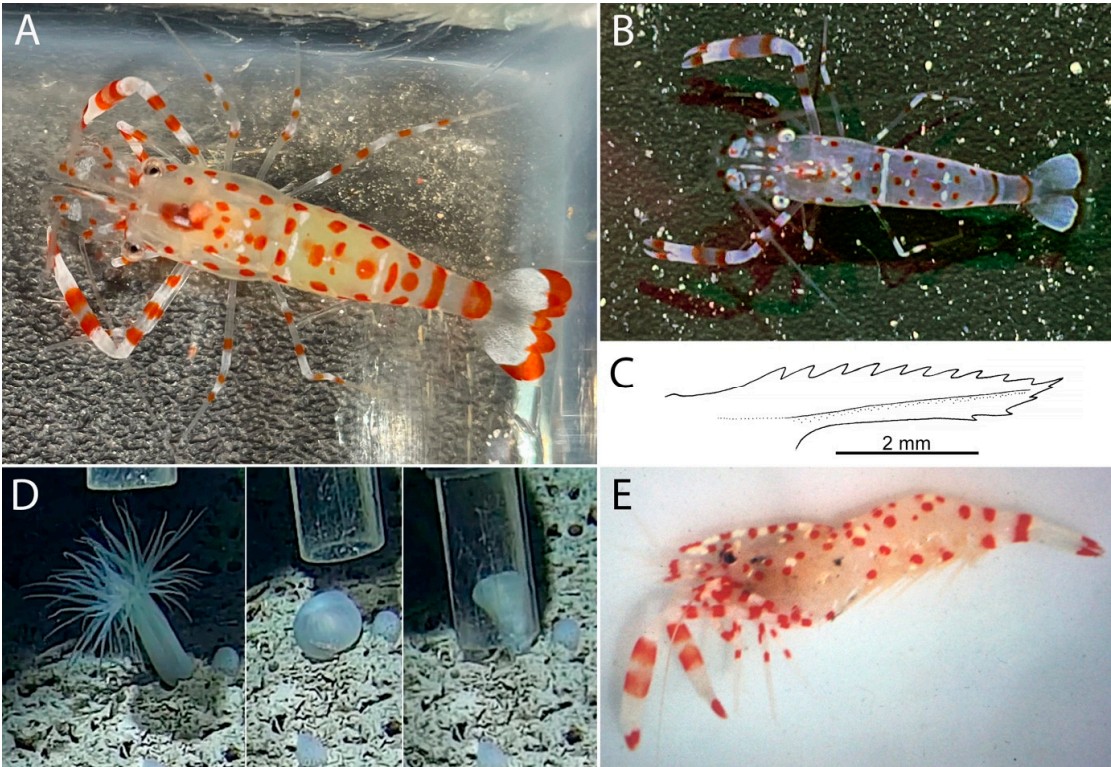

**Figure 3.** *Bruceclimenes foveolatus* (Bruce, 1981) comb. nov., ovigerous female (pocl 7.8 mm, cl 11.2 mm) from west of Umm Urumah, Saudi Arabia, FLMNH UF 68,695 (**A**); male (pocl 4.9 mm, cl 8.8 mm) from west of Mashabih, Saudi Arabia, RSRC (not deposited, see text) (**B**,**C**); unidentified sea anemone (Actiniaria), host of *B. foveolatus* (**D**); old photographic slide found in the archives of the Muséum National d'Histoire Naturelle (MNHN) showing most probably one of the type specimens from the Philippines (**E**); (**A**,**B**), shrimps alive, dorsal; (**C**), rostrum of the male in (**B**), lateral; (**D**), attempt to collect the sea anemone host, with the shrimp (male in (**B**)) silhouette in front of the retracted host in the middle panel (before suction took place); (**E**), shrimp alive, lateral. Photographs by S. Vimercati (**A**), F. Marchese (**B**) and MUSORSTOM crew/MNHN (**E**). Frame grab from a video recorded by OceanX team (**D**).

Bruce [18], in his description of *Periclimenes foresti* Bruce, 1981, another deep-water shrimp from the Philippines, stated that this species may be closely related to *P. foveolatus*. Subsequently, four other deep-water species of *Periclimenes* have been described and assigned to the *P. foresti* complex/group [47–49]. Two of these species were later transferred to *Bathymenes* [44,50]. However, the remaining two species of the *P. foresti* complex, as well as another deep-water species originally described as *Periclimenes crosnieri* Li & Bruce, 2006,

also conform to the new diagnosis of *Bathymenes* in Ďuriš & Šobáňová [50] and, therefore, must be transferred to that genus (see below). Based on overall morphology, colour patterns, available host data and preliminary molecular data (Figure 2), *Bruceclimenes* is rather distantly related to *Bathymenes* and is also not particularly closely related to *Echinopericlimenes*, another major lineage of deep-water shrimps associated with echinoderms. Several other Indo-West Pacific deep-water species of *Periclimenes s. l.*, some of them with possible affinities to *Bruceclimenes*, await generic reassignments (e.g., [22,37,51]).

### *Bruceclimenes foveolatus* (Bruce, 1981), comb. nov.
Figure 3A–C,E

*Periclimenes foveolatus* Bruce [18]: 196, Figures 6–9, 17a–b, 18b,e; Bruce [37]: 312, Figures 8, 72; Chace & Bruce [52]: 56.

*Material examined:* Saudi Arabia, Red Sea. One ovig. female (pocl 7.8 mm, cl 11.2 mm), FLMNH UF 68695, west of Umm Urumah, Red Sea Decade Expedition, Leg 3 Phase 2, sta. NTN0154, 25.770922, 36.516308, depth: 471 m, possibly on sea anemone, coll. Neptune Dive Team/S. Vimercati, 14.05.2022 (NTN0154-Bio7/OCX-117*); one male (pocl 4.9 mm, cl 8.8 mm), KAUST (not deposited, see text), west of Mashabih, Red Sea Decade Expedition, Leg 3 Discovery, sta. CHR0249, 25.654252, 36.476551, depth 321 m, on sea anemone, coll. Chimera Dive Team/F. Marchese, 19.04.2022 (CHR0249-Bio8, Slurp2/OCX-118*). *Designates sequenced specimen.

*Description*: See Bruce [18], with additional remarks and illustrations in Bruce [37].

*Colour pattern*: Body partly translucent, with yellowish tinge, covered with numerous, relatively large, rounded or ovoid, rarely more irregular, bright red spots, and less numerous, less conspicuous, smaller white spots or incomplete transverse bands; red spots smaller on carapace mid-dorsal line and flanks and larger closer to posterior margin of carapace and on pleon; white spots smaller on carapace and forming short dorsal bands on first and third pleonite; rostrum and post-rostral mid-line of carapace with conspicuous, alternating white and red markings; fifth pleonite with red transverse band dorsally, in addition to few large spots; sixth pleonite with red transverse band only, this band continues to sternal surface and thus forms a continuous ring; antennular and antennal peduncles with some white and red blotches, flagella colourless; first pereiopods semi-transparent, with transverse red and white bands distally; second pereopods (main chelipeds) white with transverse, broad, bright red bands and greyish areas between them as following: one red–greyish–red band on merus, one red–greyish–red band on proximal two-thirds of chela palm and one red band across bases of fingers; proximal and distal areas of merus, carpus, distal palm and area adjacent to fingers tips white; third to fifth pereiopods (walking legs) semi-transparent, with four transverse, broad, bright red bands, one band per article (distal part of ischium, middle of merus and carpus and proximal part of propodus), and two or three white bands (on distal merus and carpus, and sometimes, ischium); telson translucent with bright red distal third; uropods translucent proximally, white on central part and red on distal thirds of exopod and endopod (Figure 3A,B,E).

*Distribution:* Indo-West Pacific. Philippines: Ambil Island near Lubang Island (187–195 m) and southwest of Manila Bay, Luzon Island (188–191 m) [18,52]; Loyalty Islands east of New Caledonia (236 m) [37]; Red Sea: off north-central coast of Saudi Arabia (321–471 m) (present study).

*Ecology*: In the Red Sea, *B. foveolatus* associates with medium-sized (column in full extension ~15 cm, diameter with tentacles expanded ~20 cm), whitish, non-identified sea anemones (Figure 3D) at depths of 321–471 m. The identification of the host, even if it had been collected, is extremely challenging as the deep-water actiniarian fauna of the Red Sea and Indian Ocean is generally poorly known [53]. The hosts of the specimens from the Philippines and New Caledonia remain unknown.

*Remarks*: *Bruceclimenes foveolatus* was described and illustrated in detail by Bruce [18,37] as *P. foveolatus*. The male and female of *B. foveolatus* from Saudi Arabia agree well with

the original description of the species based on the material from the Philippines [18]. The foveolation on the carapace and pleon in the present material is much less marked than in the type specimens (Bruce [18]: Figure 6); however, this was also the case with the single male specimen from New Caledonia [37]. The rostral dentitions of the Saudi Arabian male and female are 10/4 and 10/3, respectively, whereas Bruce [18] gave 7–8/1–2 for the type series and 9/3 for the New Caledonian specimen; thus, the hitherto known rostral formula of *B. foveolatus* is 7–10/1–3.

The colour pattern of the Red Sea specimens (Figure 3A,B) is generally similar to that of the New Caledonian specimen, as illustrated by Bruce ([37]: Figure 72), although in the former specimens, the spots are larger and less numerous, whilst the telson is mostly colourless rather than white, as in the latter specimen. In the original description of *B. foveolatus*, Bruce ([18]: 201) stated that the colour pattern of the type specimens from the Philippines is unknown. However, a colour slide of *B. foveolatus* from the 1970s found by one of us (AA) in the slide collection of the Muséum National d'Histoire Naturelle in Paris (MNHN) presumably depicts one of the type specimens from the Philippines (Figure 3E). Remarkably, the colour pattern of this specimen is almost identical to that of the two Red Sea specimens (Figure 3A,B). Also noteworthy is the fact that in the Red Sea, *B. foveolatus* was collected in slightly deeper water, at depths of 321 m and 471 m, whereas the specimens from the Philippines and New Caledonia came from depths ranging between 187 m and 236 m.

For the time being, the slight differences in morphology and colour patterns are treated as part of intraspecific variation in a single, widely distributed, deep-water species, apparently associated with sea anemones (see above), and ranging from the Red Sea to the Philippines and New Caledonia. However, a molecular comparison between the specimens of *B. foveolatus* from these three distant localities, which presently is impossible due to a lack of fresh material from the Philippines and New Caledonia, is highly desirable.

**Michaelimenes Okuno, 2017**

*Michaelimenes* Okuno [20]: 2.

*Diagnosis*: See Okuno [20].

*Type species*: *Michaelimenes perlucidus* (Bruce, 1969), originally described as *Periclimenes perlucidus* Bruce, 1969, by original designation (synonym: *P. involens* Bruce, 1996).

Other species included: Michaelimenes latipollex (Kemp, 1922); Michaelimenes platydactylus (Li, 2008); Michaelimenes sammy sp. nov.; Michaelimenes wirtzi (d'Udekem d'Acoz, 1996) comb. nov., originally described as Periclimenes wirtzi d'Udekem d'Acoz, 1996.

*Hosts*: Soft corals (Alcyonaria) and black corals (Antipatharia).

*Bathymetric range*: 15–494 m, but mainly in the depth range of 20–200 m.

*Distribution*: Indo-West Pacific from the Red Sea to Japan and French Polynesia [16,20,54,55], present study; East Atlantic from Madeira and Azores to the Cape Verde Islands [56,57].

*Remarks*: In the generic diagnosis of *Michaelimenes* Okuno, 2017, [20] stated that the pollex (=fixed finger) of the second pereiopod chela is proximally excavated, whereas the dactylus is characterised by a dorsal flange. All species currently assigned to *Periclimenes* that present only the second condition, i.e., a flanged dactylus on the second pereiopod, were excluded from *Michaelimenes* by Okuno [20]. These species are *P. carinidactylus* Bruce, 1969 from southern Australia [58,59]; *P. compressus* Borradaile, 1915 from the central Indian Ocean [60,61]; *P. tenellus* (Smith, 1882) from the western Atlantic [62–64]; and *P. infraspinis* (Rathbun, 1902) from the eastern Pacific [64,65]. Furthermore, *P. forcipulatus* Bruce 1991 from the Loyalty Islands near New Caledonia [37] was discussed by Okuno [20] but excluded from *Michaelimenes* because of the presence of minute teeth on the fingers of the first pereiopod in this species. Thus, *Michaelimenes* hitherto included only three species, viz. the type species *M. perlucidus* (Bruce, 1969), *M. latipollex* (Kemp, 1922), both with several records from across the Indo-West Pacific; and *M. platydactylus* (Li, 2008) from French Polynesia (see [16,20,55,59] and references therein).

Borradaile [60] and Kemp [16] considered *P. laccadivensis* Alcock & Anderson, 1894 as possibly close to *M. latipollex* (see also [37,66]). However, in *P. laccadivensis*, the hepatic tooth is in a much lower position than in *M. latipollex* ([16,67], see also Bruce [37]: Figure 1a,b) and the below-described *M. sammy* (Figure 4B), whereas the dactylus of the second pereiopod is not flanged, although the pollex is shallowly excavated in its proximal part (Kemp [16]: Figure 20b). The lateral flange on the dactylus of the second pereiopod is also present in many species of *Bathymenes*, which differ from all the above-mentioned species by the presence of conical or subacute granules on the second pereiopods (at least on the chelae), and several other features [44,50]. In the molecular analyses, *P. laccadivensis* was recovered either as a sister of *B. boucheti* in Gan et al. [21] or in an unresolved position within Clade 7 in Horká et al. [8]. In our molecular analysis (Figure 2), *P. laccadivensis* clusters with two species of *Echinopericlimenes*, although with relatively low branch support. Whatever its true affinities might be, *P. laccadivensis* does not seem to be closely related to *Michaelimenes* and is therefore not included in that genus.

On the other hand, *M. sammy* and the eastern Atlantic *P. wirtzi* have numerous morphological similarities, including the dactylar flange in the second pereiopod chela and what can be interpreted as a shallow proximal excavation on the pollex (cf. d'Udekem d'Acoz [56]: Figure 7), share the same host group, i.e., Antipatharia ([68]; see also insert in Figure 2) and were recovered in sister position in our molecular analysis, although with a significant genetic divergence (Figure 2). Therefore, *P. wirtzi* is transferred to *Michaelimenes* as *Michaelimenes wirtzi* (d'Udekem d'Acoz, 1996) comb. nov. It must be noted, however, that the type species of *Michaelimenes*, *M. perlucidus*, lives on soft corals (Alcyonaria), but as pointed out by several previous studies, including Marin & Anker [69] and Horká et al. [8], host switches are very common in palaemonid shrimps. The only substantial deviation in *M. wirtzi* from the generic diagnosis of *Michaelimenes* in Okuno [20] appears to be the shape of the rostrum, which is proximally arched, then descending and ascendant again towards the tip in *M. wirtzi* (d'Udekem d'Acoz [56]: Figure 2), whilst being more or less horizontal in the four Indo-West Pacific taxa (Figure 3B; Okuno [20]: Figure 3A).

*Michaelimenes sammy* **sp. nov.** (authored by Anker)
(urn:lsid:zoobank.org:act:8945277C-47BE-475E-B1E3-9E34ACB7D59C)
Figures 4, 5 and 6A,B,D

*Type material*: Saudi Arabia, Red Sea. Holotype: ovig. female (pocl 3.2 mm, cl 5.9 mm), FLMNH UF 68694, north of small islet ~25 km west of Jabal Alsabaya, Red Sea Decade Expedition, Leg 2, Phase 2, sta. NTN0133, 18.514100–18.510279, 40.666581–40.665597, max. depth 302 m, coll. Neptune Dive Team/F. Benzoni and F. Barreca, 01.04.2022 [OCX-037]. Paratypes: one female (pocl 2.1 mm, cl 3.9 mm), one ovig. female (pocl 2.8 mm, cl 5.6 mm), FLMNH UF 68692, same collection data as for the holotype (OCX-035*); one female (pocl 1.5 mm, cl 2.6 mm), KAUST, north of small islet 25 km west of Jabal Alsabaya, Red Sea Decade Expedition, Leg 2, Phase 2, sta. NTN0134, 18.812696, 40.637453, depth 191 m, coll. Neptune Dive Team/F. Benzoni and F. Barreca, 02.04.2022 (NTN0134-bycatch/OCX-104*); one ovig. female (pocl 2.9 mm, cl 5.3 mm), FLMNH UF 68693, south of Jazirt Abu Rashed, Neom, Deep Blue Expedition, sta. NTN0029, 27.600706, 35.334426, depth 494.1 m, coll. Neptune Dive Team/F. Marchese, 10.10.2020 (OCX-002*); one male (pocl 3.1 mm, cl 6.2 mm), RSRC (not deposited, see text), off Al Muwaileh, Neom, OceanX Red Sea Relationship Cultivation, sta. CHR0310, 27.632867, 35.311688, depth 407 m, coll. Chimaera Dive Team/F. Benzoni, F. Marchese and T.I. Terraneo, 23 June 2022 (CHR0310-8B/OCX-210); one ovig. female (pocl 2.3 mm, cl 4.4 mm), KAUST (not deposited, see text), north-west of Jeddah, 10 km west of Nelson Reef, Red Sea Decade Expedition, Leg 1, Phase 2, sta. NTN0121, 21.90876, 38.76103, depth 91.3 m, on black coral, coll. Neptune Dive Team/T.I. Terraneo, 22.02.2022 (NTN0121-Bio19B/OCX-202). *Designates sequenced specimen.

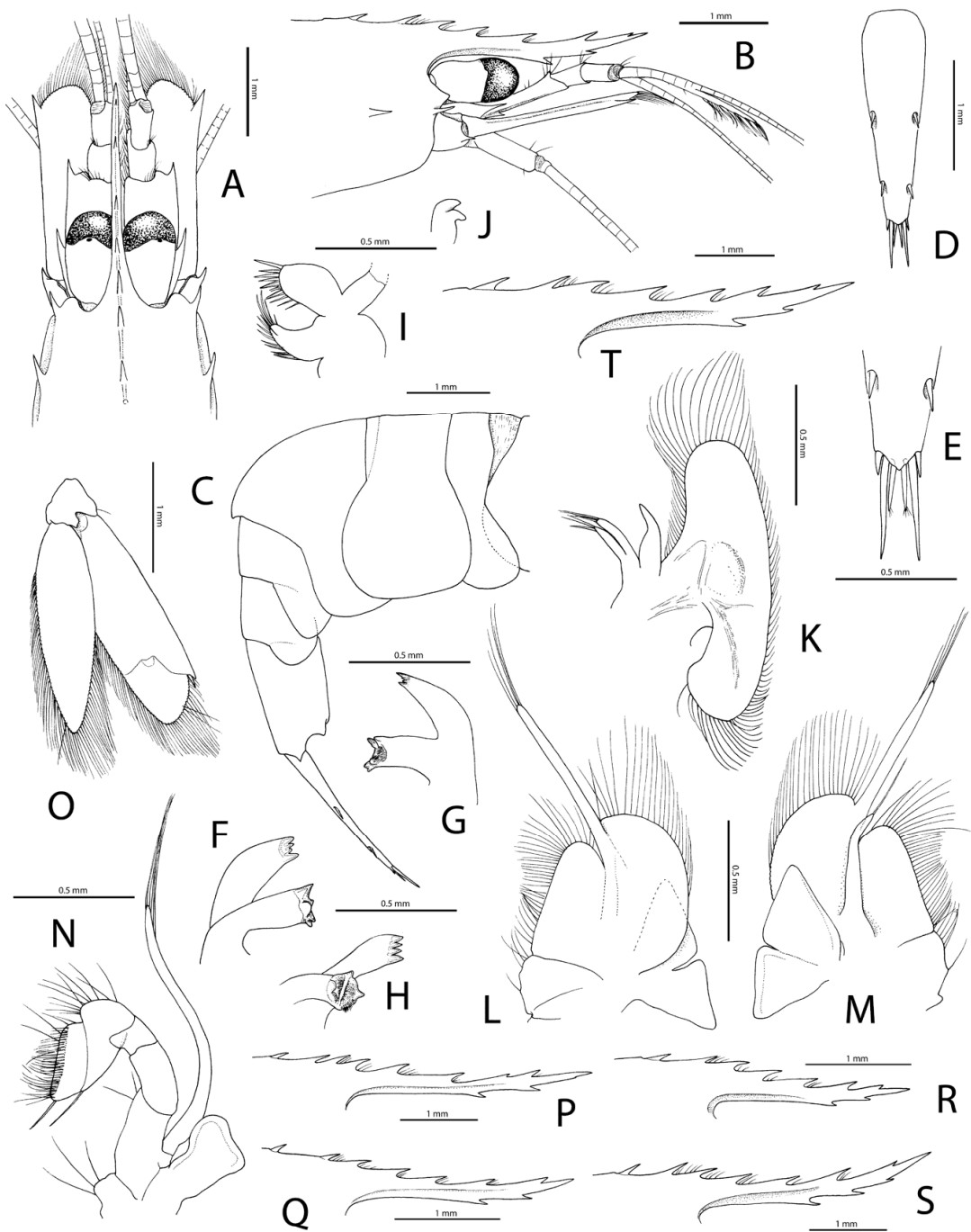

**Figure 4.** *Michaelimenes sammy* sp. nov., paratype, ovigerous female (pocl 2.9 mm, cl 5.3 mm) from south of Jazirt Abu Rashed, Saudi Arabia, FLMNH UF 68,693 (**A–O**); paratype, male (pocl 3.1 mm, cl 6.2 mm) from off Muwaileh, Saudi Arabia, RSRC (not deposited, see text) (**P**); paratype, ovigerous female (pocl 2.3 mm, cl 4.4 mm) from northwest of Jeddah, RSRC (not deposited, see text) (**Q**); paratypes, two females, including one ovigerous (pocl 2.1 mm, cl 3.9 mm; pocl 2.8 mm, cl 5.6 mm), from west of Jabal Alsabaya, Saudi Arabia, FLMNH UF 68,692 (**R,S**); holotype, ovigerous female (pocl 3.2 mm, cl 5.9 mm) from the same locality, FLMNH UF 68,694 (**T**); (**A**), frontal region, dorsal; (**B**), same, lateral; (**C**), pleon, lateral; (**D**), telson, dorsal; (**E**), same, posterior third, dorsal; (**F**), mandible, mesial; (**G**), same, lateral; (**H**), same, anterior; (**I**), maxillule, lateral; (**J**), same, detail of palp (seta of ventral lobe possibly broken off); (**K**), maxilla, lateral; (**L**), first maxilliped, lateral; (**M**), same, mesial; (**N**), second maxilliped, lateral; (**O**), uropod, dorsal; (**P–T**), rostra of different specimens, including holotype (**T**), lateral.

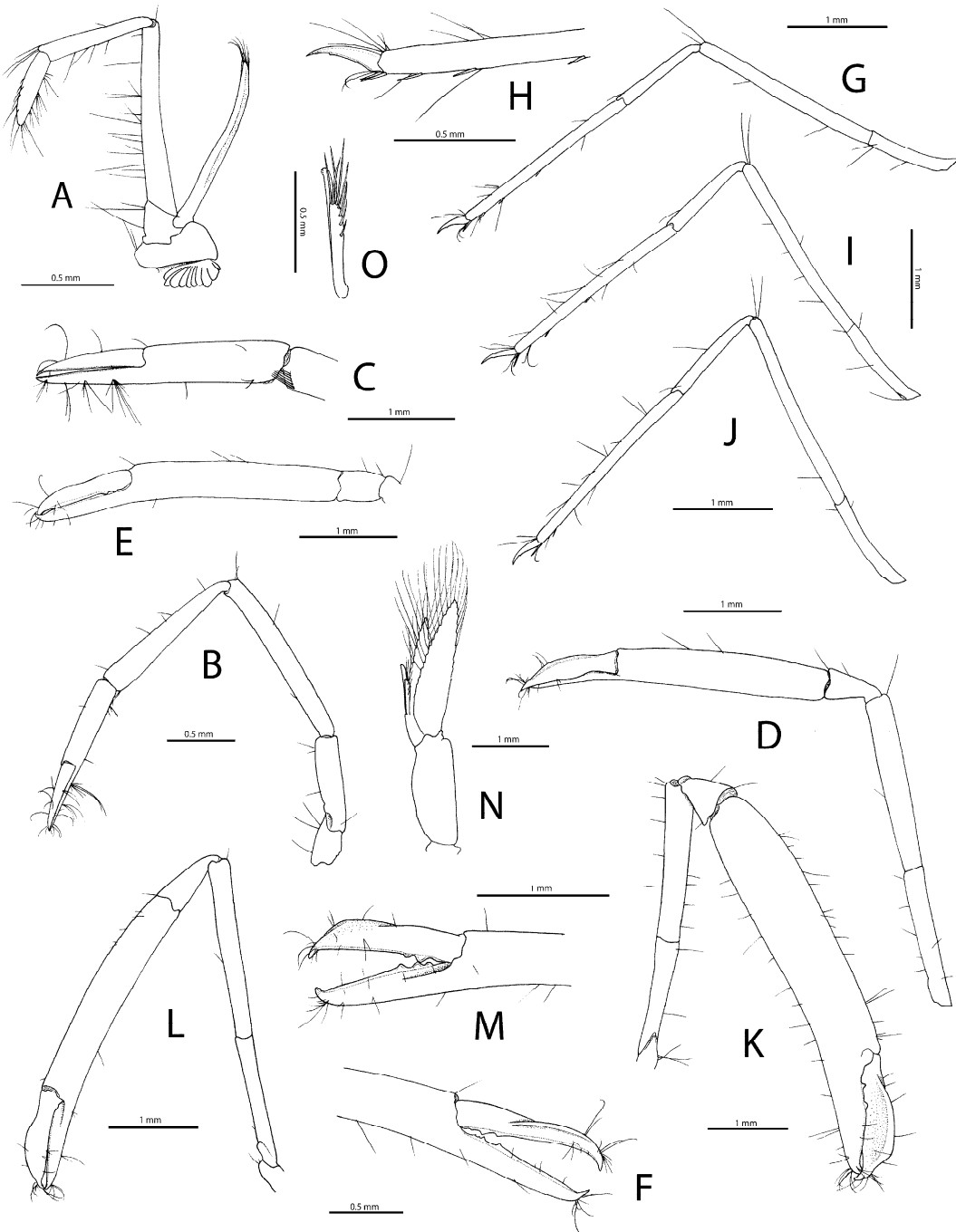

**Figure 5.** *Michaelimenes sammy* sp. nov., paratype, ovigerous female (pocl 2.9 mm, cl 5.3 mm) from south of Jazirt Abu Rashed, Saudi Arabia, FLMNH UF 68,693 (**A–J**); holotype, ovigerous female (pocl 3.2 mm, cl 5.9 mm) from west of Jabal Alsabaya, Saudi Arabia, FLMNH UF 68,694 (**K**); paratype, male (pocl 3.1 mm, cl 6.2 mm) from off Muwaileh, Saudi Arabia, RSRC (not deposited, see text) (**L–O**); (**A**), third maxilliped, lateral (arthrobranch somewhat displaced); (**B**), first pereiopod, lateral; (**C**), same, distal part of carpus and chela, lateral; (**D**), left (major) second pereiopod, lateral; (**E**), same, chela and carpus, lateral; (**F**), same, distal part of palm and fingers open, mesial; (**G**), third pereiopod, lateral; (**H**), same, distal half of propodus and dactylus, lateral; (**I**), fourth pereiopod, lateral; (**J**), fifth pereiopod, lateral; (**K**), right (major) second pereiopod of female holotype (drawn *in situ*), lateral; (**L**), left (major) second pereiopod of male paratype (drawn *in situ*), lateral; (**M**), same, distal part of palm and fingers open, lateral; (**N**), male second pleopod, anterior; (**O**), same, detail or appendices masculina and interna, anterior.

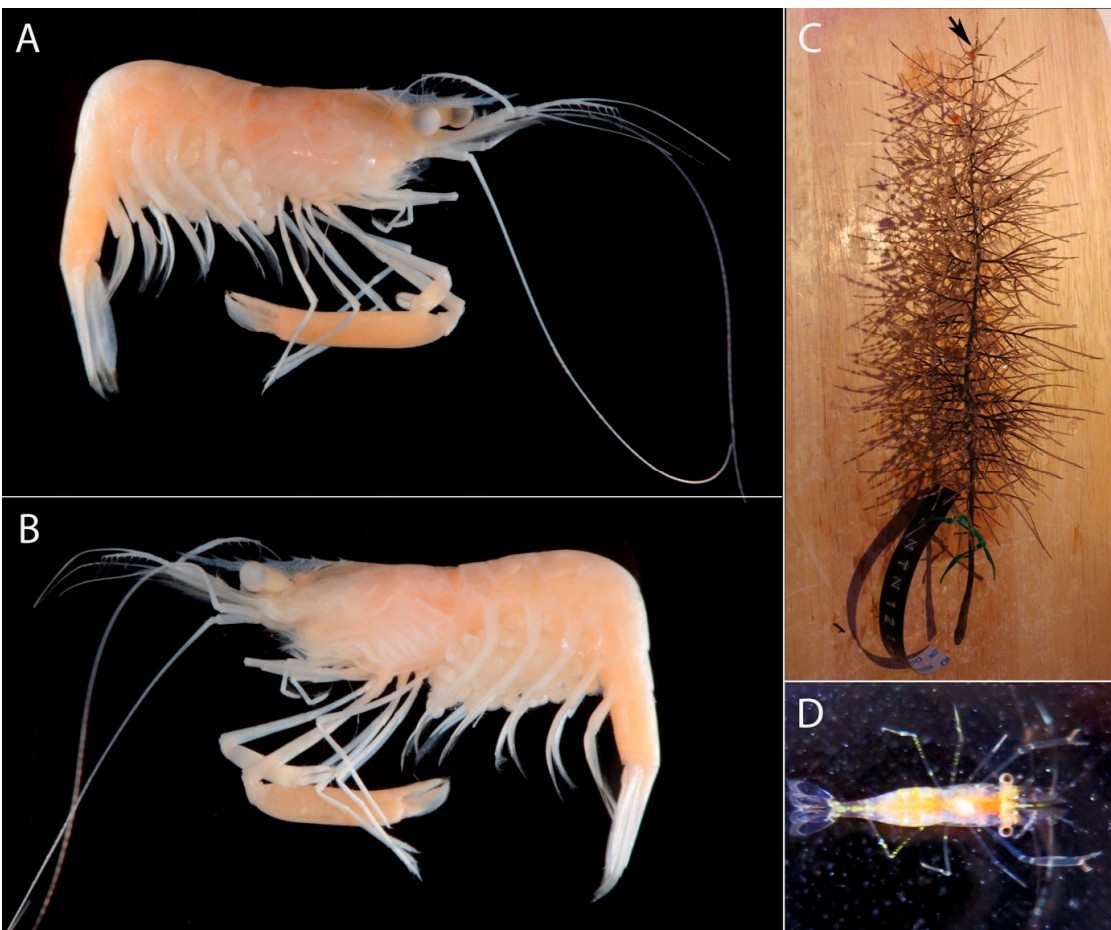

**Figure 6.** *Michaelimenes sammy* sp. nov., holotype, ovigerous female (pocl 3.2 mm, cl 5.9 mm) from off Jabal Alsabaya, Saudi Arabia, FLMNH UF 68,694 (**A**,**B**); black coral of the family Myriopathidae, host of following paratype specimen (C); paratype, ovigerous female (pocl 2.3 mm, cl 4.4 mm) from north-west of Jeddah, Saudi Arabia, RSRC (not deposited, see text) (**D**); (**A**,**B**), preserved specimen, right lateral (**A**), left lateral (**B**); (**C**), black coral host shortly after collection, with shrimp clinging to one of the upper branches (indicated with a black arrow); (**D**), shrimp alive, dorsal. Photographs by A. Anker (**A**,**B**) and T.I. Terraneo (**C**,**D**).

*Description*: Small-sized palaemonid shrimp (type material: pocl 1.5–3.2 mm) with cylindrical, not noticeably depressed or compressed body. Carapace smooth, glabrous. Rostrum (Figure 4A,B,P–T; see also Figure 6A,B) moderately high, straight, horizontal or shallowly concave near mid-length and ascendant towards tip, overreaching level of distal end of antennular peduncle, about as long or slightly shorter than carapace (postorbital length); dorsal margin armed with six to seven teeth, posterior-most tooth (rarely two teeth) situated posterior to or above posterior orbital margin; ventral margin straight or slightly convex, armed with two to three teeth on distal half; lateral carinae well marked; supraorbital eaves absent. Orbit (Figure 4A,B) feebly developed; inferior orbital angle bluntly produced anteriorly. Antennal tooth (Figure 4A,B) well developed, submarginal, sharp, overreaching level of distal end of inferior orbital angle. Hepatic tooth strong, sharp, situated posterior to and almost at same level as antennal tooth (Figure 4A,B). Epigastric tooth (Figure 4A,B,P–T) present, well-developed (missing, possibly broken off, in smallest specimen), somewhat removed from posterior-most tooth of rostral series, situated well posterior to the level of posterior orbital margin and slightly posterior to the level of hepatic tooth, semi-mobile, separated from carapace by well-demarcated suture. Supraorbital tooth absent. Ophthalmic somite without pronounced bec ocellaire (=interocular beak). Thoracic sternite 4 lacking median process. Epistome unarmed.

Pleon (Figure 4C) smooth, glabrous, with all pleura rounded ventrally; pleonite 3 with posterior margin slightly produced, overhanging dorsal surface of pleonite 4, forming low hump; pleonite 6 not particularly elongate, 1.5 times as long as high, shorter than telson. Telson (Figure 4D,E) narrow, gradually tapering towards posterior end; dorsal surface with two pairs of stout, submarginal spiniform setae located near mid-length and 0.8 of telson length, respectively; posterior margin subtriangular, with three pairs of spiniform setae, lateral ones stout, short, about 0.4 length of slender median ones and about 0.25 length of similarly stout intermediate ones.

Antennule (Figure 4A,B) with first article of peduncle robust, much longer than the length of second and third articles combined; lateral margin somewhat convex, terminating distally in strong acute tooth; distal margin slightly convex mesial to distolateral tooth; stylocerite small, acute, overreaching 0.4 length of first article; ventromesial margin with small acute tooth; second and third articles subequal in length, much slenderer than first, unarmed; lateral flagellum biramous; accessory free ramus well developed, as long as fusion portion, but with poorly defined subdivisions, with numerous groups of aesthetascs. Antenna (Figure 4A,B) with basicerite small, short, distally armed with sharp tooth; scapho-cerite well developed; lateral margin straight, terminating in stout, sharp distolateral tooth, the latter not reaching bluntly angular distal margin of blade; carpocerite very short, not reaching 0.4 length of scaphocerite; flagellum normal, not thickened.

Mouthparts not especially modified, as illustrated (Figures 4F–N and 5A). Mandible (Figure 4F–H) without palp; molar process well developed; incisor process with four teeth distally. Maxillule, maxilla and maxillipeds 1 and 2 (Figure 4I–N) without diagnostic features, as illustrated. Maxilliped 3 (Figure 5A) with coxa bearing semicircular lateral plate; endopod moderately slender and setose; antepenultimate article clearly separated from basis, without spiniform setae; penultimate article about half as long as antepenultimate article, distally widening; ultimate article with stiff, non-spiniform setae; exopod short, reaching about 0.8 length of antepenultimate article; arthrobranch well developed.

Pereiopod 1 (Figure 5B,C) slender, moderately elongate; ischium about half-length of merus; merus and carpus subequal in length; carpo-propodal grooming apparatus mod-estly developed; chela about 0.7 length of propodus; fingers about 0.7 length of propodus; dactylus not subspatulate; cutting edges of both fingers unarmed. Pereiopods 2 unequal in length, slightly dissimilar in shape and proportions, elongate, smooth (Figures 5D–F,K–M and 6A,B). Major pereiopod 2 (Figure 5D–F,K–M) with coxa lacking marked ventral pro-cess; basis very short; ischium slender, widening, markedly shorter than merus; merus more or less widening distally, unarmed; carpus variable in length and shape depend-ing on age of individuals, short and cup-shaped in large specimens (Figure 5K), more elongate in smaller specimens (Figure 5L), always widening distally; chela feebly to moder-ately enlarged depending on age of individuals, particularly robust in larger specimens (Figures 5K and 6A,B); palm cylindrical in cross-section, 4–5 times as long as high, smooth, sparsely setose; fingers 0.4–0.5 length of palm, slightly gaping distally, with crossing finger-tips; dactylus with conspicuous lateral flange, especially well-marked in large specimens; pollex with shallow excavation proximally; cutting edge of each finger armed with two to three rounded or subacute teeth on proximal third of finger length. Minor pereiopod 2 variously smaller and weaker than major pereiopod 2 (Figure 6A,B), generally similar in shape; dactylar flange much less obvious than in major pereiopod 2.

Ambulatory pereiopods (P3–5) similar, slender, elongate (Figure 5G–J); ischium about 0.4 length of merus; merus about 14 (P3) to 15 (P5) times as long as wide, about twice as long as carpus; propodus subequal to merus in length; ventral margin of P3 and P4 with 3 spiniform setae on distal half and 1 pair of spiniform setae adjacent to dactylar base; ventromesial margin of P5 with 1 slender spiniform seta at about 0.9 length of propodus and one longer and stouter distal spiniform seta adjacent to dactylar base; propodal cleaning brush not distinct; dactylus less than 0.2 length of propodus, simple, moderately stout, subtly biunguiculate, with minute secondary unguis (sometimes notch) on stout corpus and slender, distinctly curved terminal unguis.

Male first pleopod without notable features. Male second pleopod (Figure 5N,O) with endopod bearing typical appendices masculina and interna, former not reaching tip of latter, with stiff setae on apex and along distal half of mesial margin. Uropod (Figure 4O) exceeding telson; protopod with two distally blunt lobes; exopod ovate-rectangular, rounded distally; distolateral tooth smooth; adjacent spiniform seta moderately strong; endopod narrowly ovoid, slightly shorter than exopod, its distal margin reaching to about 0.9 length of exopod.

*Colour pattern*: Translucent with pale orange tinge; ventral side of cephalothorax and pleon reddish; second pereiopods translucent with pale reddish fingers; walking legs with yellow spots; tail fan largely translucent (Figure 6D).

*Etymology*: The new species is named after our colleague, Sammy De Grave (Oxford University Museum of Natural History), a well-known expert on caridean shrimps with a slight fondness for symbiotic palaemonid shrimps. Sammy's first (given) name is used as a noun in apposition.

*Distribution*: Presently known only from northern and central Red Sea, off the coast of Saudi Arabia.

*Ecology*: The recorded depth range of the type series is 91.3–494.1 m; one of the specimens was found on a colony of bushy black corals of the family Myriopathidae (Figure 6C) indicating a possible association between *M. sammy* and this group of cnidarians.

*Remarks:* The overall combination of morphological characteristics strongly suggests that *M. sammy* is a member of *Michaelimenes*, displaying all essential diagnostic features of the genus (Figures 4 and 5; Okuno [20]). The proximal excavation on the cutting edge of the second pereiopod pollex is relatively shallow and best visible on the major chela of the largest specimens (Figure 5K; see also below). *Michaelimenes sammy* can be easily separated from both *M. perlucidus* and *M. latipollex* by the obscurely biunguiculate dactylus of the third to fifth pereiopods (*vs.* strongly biunguiculate in *M. perlucidus* and *M. latipollex*); from *M. perlucidus* by the noticeably larger distance between the epigastric tooth and the posterior-most rostral tooth; from *M. latipollex* by the much more slenderer dactylus of the third to fifth pereiopods (in addition not being strongly biunguiculate, see above); and from *M. platydactylus* by the posterior-most tooth of the dorsal rostral series in epigastric position, as in *M. latipollex* (*vs.* situated above the orbit in *M. platydactylus*), and the noticeably slenderer third pereiopod (cf. Kemp [16], Okuno [20] and Li [55]). In the position of the hepatic tooth, *M. sammy* is most similar to *M. latipollex*; in *M. perlucidus* and *M. platydactylus*, the hepatic tooth is closer to and slightly lower than the antennal tooth (Figure 4B; cf. Kemp [16], Okuno [20] and Li [55]). On the other hand, in the general shape of the dactylus of the walking legs, *M. sammy* is closer to *M. platydactylus*.

The separation of *M. sammy* from the eastern Atlantic *M. wirtzi* is relatively straightforward. The two species greatly diverge in the shape and length of the rostrum, which in the new species is usually straight (horizontal) or at most slightly concave and then ascendant, armed with two or three teeth ventrally, and with the tip only slightly overreaching the distal end of the antennular peduncle (*vs.* proximally strongly arched and then concave all the way to the tip, armed with five to seven teeth on the ventral margin in the adults, and with the tip by far overreaching the distal end of the antennular peduncle in *M. wirtzi*) (Figure 4B,P–T; cf. d'Udekem d'Acoz [56]: Figures 1 and 2A, D–H). They also differ in the proportions of the penultimate article of the third maxilliped and the degree of convexity of the distal margin of the antennal scaphocerite blade (Figures 4 and 5A; cf. d'Udekem d'Acoz [56]: Figures 3A and 4D).

The excavation of the pollex of the second pereiopod chela, which was considered by Okuno [20] as one of the most important diagnostic characters of *Michaelimenes*, is quite deep in *M. perlucidus*, at least as illustrated by Okuno ([20]: Figure 1B,E), but may be shallower and more subtle in other species, including *M. sammy* (Figure 5K). Kemp ([16]: 151) described the cutting-edge area of the second pereiopod chela as "the single tooth on the dactylus fitting into a recess in the fixed finger", whereas Li ([55]: 233) stated that the "fixed finger length larger than dactylar teeth, with proximal excavation present to

fit proximal part of dactyl cutting edge", although he did not show it in his illustration of the opened chela fingers (cf. [56]: Figure 19D). In fact, the presence or absence of a proximal excavation on the pollex of the second pereiopod chela alone should not exclude at least some species of *Periclimenes* from *Michaelimenes*. Indeed, as noted by Okuno [20], the current taxonomic status of four species of *Periclimenes* characterised by the flanged dactylus of the second pereiopod chela, but apparently without a proximal excavation on the pollex, viz. *P. carinidactylus*, *P. compressus*, *P. tenellus* and *P. infraspinis*, may need to be reassessed in the future. This is especially true for *P. tenellus*, which possesses at least a shallow depression on the pollex of the second pereiopod chela [63,64] and may well represent a western Atlantic species of *Michaelimenes*. In the absence of DNA data for these taxa, it is more prudent not to transfer *P. tenellus* and the remaining three species from *Periclimenes* to *Michaelimenes*. Nevertheless, they need to be considered in the comparison with the herein-described new species of *Michaelimenes*.

*Michaelimenes sammy* can be easily separated from *P. carinidactylus* by the submarginal position of the dorsal spiniform setae of the telson (*vs.* clearly marginal in *P. carinidactylus*), the hepatic tooth situated at the same level as the antennal tooth (*vs.* noticeably lower in *P. carinidactylus*), the posterior-most tooth in the dorsal rostral series in epigastric position (*vs.* much more anterior in *P. carinidactylus*) and the slenderer dactylus of the third to fifth pereiopods (*vs.* much stouter in *P. carinidactylus*) (Figure 5F–I; cf. Bruce [59]: Figures 2A,F and 4K); from *P. compressus* by two features on the third and fourth pereiopods, namely, their propodi armed with spiniform setae distally and their dactyli longer and less curved, and the first pereiopod carpus much longer than the chela (*vs.* distinctly shorter than the chela in *P. compressus*) (Figure 5F–H; cf. Bruce [61]: Figure 4G); from *P. tenellus* by the hepatic tooth as large as and at the same level as the antennal tooth, also being in a more posterior position (*vs.* much larger and lower than the antennal tooth and much closer to it in *P. tenellus*), and by the obscurely biunguiculate dactylus of the third to fifth pereiopods (*vs.* strongly biunguiculate in *P. tenellus*) (Figures 4B and 5F–H; cf. Anker et al. [64]: Figure 3b, h); and finally, from *P. infraspinis* by the hepatic tooth as large as and at the same level as the antennal tooth (*vs.* much lower in *P. infraspinis*), the dorsally non-elevated, more or less horizontal rostrum (*vs.* dorsally strongly convex in *P. infraspinis*), and the subtly biunguiculate dactyli on the third to fifth pereiopods (*vs.* strongly biunguiculate in *P. infraspinis*) (Figures 4B and 5F,G; cf. Holthuis [63]: pl. 13, figs. a, k).

*Rubiyana* **gen. nov.** (authored by Anker)
(urn:lsid:zoobank.org:act:85F38AF2-5452-4C32-A681-994245DB129C)

*Diagnosis*: Small-sized palaemonid shrimps (pocl 1.7–3.2 mm). Body subcylindrical, neither strongly compressed not depressed, smooth. Rostrum well developed, high, horizontal, reaching or almost reaching end of antennular peduncle; dorsal margin straight, with eight to nine teeth; ventral margin almost straight, with two to three teeth in distal half; rostral carina moderate, not extending beyond posterior-most tooth. Inferior orbital angle strongly anteriorly produced, not reaching beyond antennal tooth, blunt. Antennal and hepatic teeth well-developed, acute; latter stouter and situated slightly below former, not reaching anterolateral margin of carapace. Epigastric tooth present, with or without basal suture. Supraorbital teeth absent. Thoracic sternite 4 without median process. Pleonite 3 not carinate, slightly posteriorly produced, forming very low hump; pleura of pleonites 1–5 posteroventrally rounded; pleonite 6 elongate. Telson with two pairs of dorsal spiniform setae in posterior half, inserted at some distance from margin, and three pairs of short spiniform setae on posterior margin, mesial ones with secondary setules; posterior margin bluntly triangular. Antennule typical for family; stylocerite short, acute; distolateral tooth of first article of peduncle sharp; lateral flagellum biramous, with short fused basal part; secondary ramus poorly developed. Antenna typical for family; scaphocerite well developed, with broad blade and stout distolateral tooth falling short of anterior margin of blade; carpocerite short. Eyes well developed, with large globular cornea. Mandible without palp; molar and incisor processes normal. Maxillule with deeply bilobed palp (endopod),

ventral lobe strongly hooked distally. Maxilla with reduced coxal endite and bilobed basial endite; palp (endopod) large; scaphognathite broad. Maxilliped 1 with endites fused; palp (endopod) small, entire; exopod long, with well-developed caridean lobe; epipod small, subtriangular. Maxilliped 2 with normal endopod and exopod; epipod small, ovate, without podobranch. Maxilliped 3 with coxa bearing broad, semicircular, lateral plate; endopod moderately slender; ischiomerus (antepenultimate article) not clearly separated from basis, without spiniform setae; ultimate article with robust, but non-spiniform setae; exopod short; arthrobranch absent. Pereiopod 1 slender; carpus more or less elongate; carpo-propodal brush present; chela not elongate, not inflated, with fingers as long as or shorter than palm; cutting edges of fingers unarmed. Pereiopods 2 moderately stout, similar in shape, slightly unequal in size, smooth; merus and carpus unarmed; carpus more or less elongate; chela not particularly swollen; palm subcylindrical, moderately elongate; fingers much shorter than palm; cutting edge of pollex and dactylus each armed with small tooth at proximal 0.3 of length; tooth–fossa snapping mechanism lacking. Pereiopods 3–5 slender; propodus with slender spiniform setae on ventral margin; dactylus simple. Male pleopod 1 and 2 unknown. Uropod with protopod with 2 bluntly ending lobes; exopod unarmed, except for slender, distolateral spiniform seta; endopod narrower and significantly shorter than exopod, not overreaching 0.75 of exopod length.

*Type species*: *Rubiyana arabica* sp. nov. (authored by Anker), by present designation and monotypy.

*Other species included*: None (but see remarks on *Periclimenes kallisto* Bruce, 2008 below).

*Etymology*: The name of the new genus derives from the Arabic word "rubiyan", for shrimp or prawn. Gender feminine.

*Hosts*: Soft corals (Malacalcyonacea: Nephtheidae).

*Remarks*: Our molecular analysis recovered the type species of *Rubiyana* very distantly from *Periclimenes s. str.*, more precisely as a sister to a small clade composed of *Ancylocaris brevicarpalis*, *Periclimenes affinis* and *P. kallisto* (Figure 2). A phylogenetic affinity between *P. affinis* and *A. brevicarpalis* was first indicated by Gan et al. [21], whose molecular analysis did not include *P. kallisto*. Horká et al. [8], using sequences of *P. affinis* from Gan et al. [21] and adding their sequences of *P. kallisto*, recovered a clade composed of *A. brevicarpalis*, *P. affinis* and *P. kallisto*, with the same topology as in our analysis (Figure 2; [8]: Figures 2 and 3). In the revalidation of *Ancylocaris*, Ďuriš & Horká [10] mentioned some morphological characters shared by *A. brevicarpalis*, *P. affinis* and *R. kallisto*, and also between *A. brevicarpalis* and *P. affinis*, and between *P. affinis* and *R. kallisto*. With the addition of *Rubiyana*, this lineage is hereafter called the *Rubiyana–Ancylocaris* clade (Figure 2). In addition, Bruce [19] stated that two further species of *Periclimenes* may be part of the same species group as *P. affinis* and *R. kallisto*, more precisely, *P. canalinsulae* Bruce & Coombes, 1997 and *P. jugalis* Holthuis, 1952.

The type species of *Rubiyana* is remarkably similar to *P. kallisto*. In both species, the uropodal endopod is markedly reduced in size and length, reaching only 0.75 length of the exopod (Figure 7O; Bruce [19]: Figure 2I). In other words, in both species, the distal margin of the uropodal endopod does not exceed the level of the distolateral tooth of the uropodal exopod. The greatly shortened uropodal endopod is one of the most diagnostic features of *Rubiyana*, and the presence of this feature in *P. kallisto*, but not in *A. brevicarpalis* (Lee & Koh [70]: Figure 2L), is somewhat intriguing given the obtained topology (Figure 2). The relative size and length of the uropodal endopod and exopod were neither described nor illustrated for *P. affinis* [59,71] or *P. jugalis* [72]. Bruce [59] only noted that in *P. affinis*, the "uropods present no specific features", whilst Holthuis [72] stated that in *P. jugalis*, the "uropods reach beyond the end of the telson, the endopod, however, fails to reach the end of the posterior spinules of the telson", suggesting that the endopod is only slightly shorter than the exopod, as in *P. canalinsulae* (Bruce & Coombes [73]: Figure 3N). The present topology (Figure 2) suggests that the unusual configuration of the uropod in *R. arabica* and *P. kallisto* may be the result of convergent evolution. Therefore, *P. kallisto*, *P. affinis*, *P. jugalis* and *P. canalinsulae* are not included in *Rubiyana*, awaiting these and perhaps further species of the *Rubiyana–Ancylocaris* clade to be described in more detail and genetically analysed.

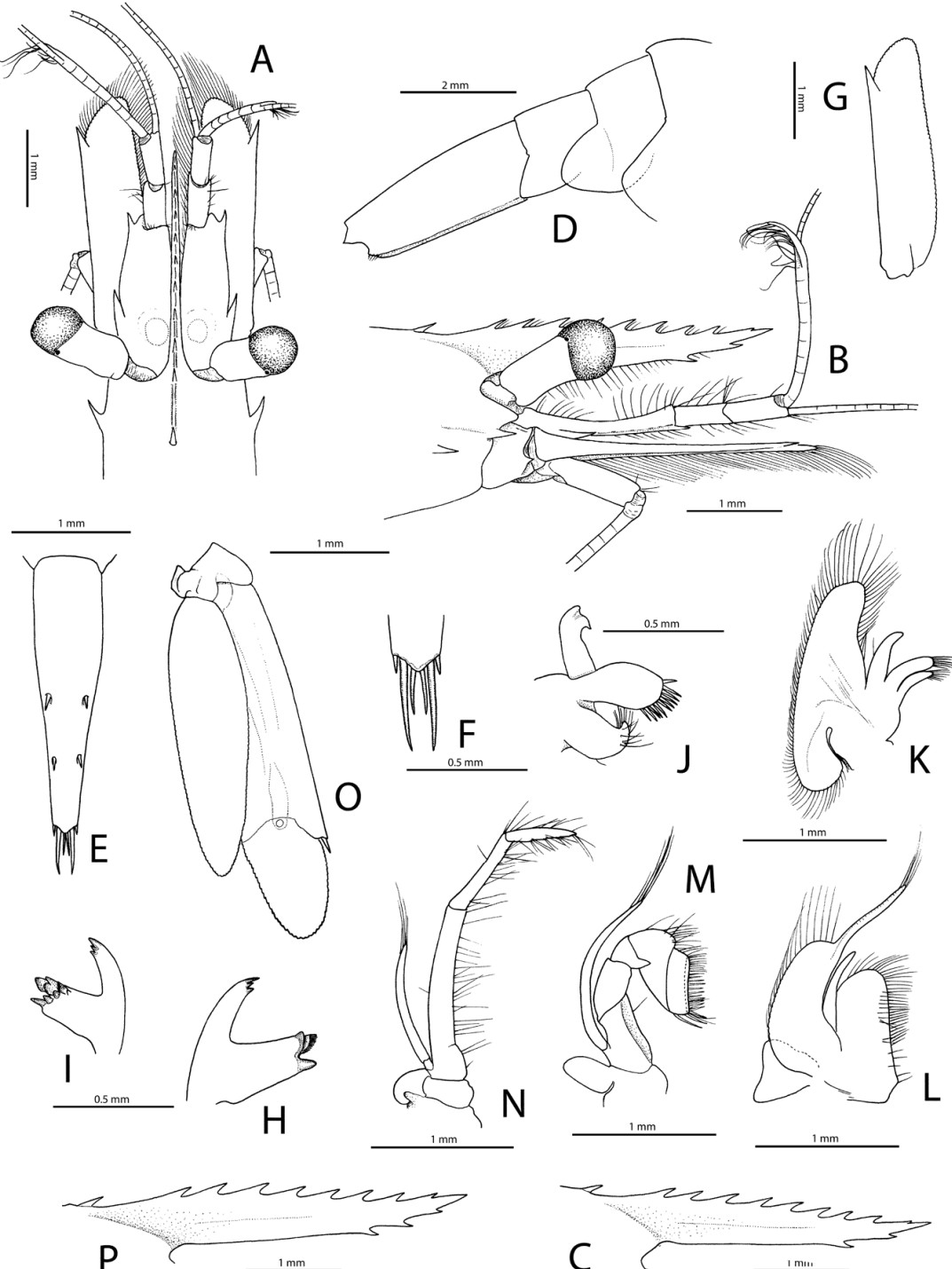

**Figure 7.** *Rubiyana arabica* gen. et sp. nov., paratype, ovigerous female (pocl 2.9 mm, cl 6.0 mm) from off Jeddah, Saudi Arabia, RSRC (not deposited, see text) (**A–O**); holotype, ovigerous female (pocl 3.2 mm, cl 6.5 mm) from northwest of the Farasan Islands, Saudi Arabia, RSRC (not deposited, see text) (**P**); A, frontal region, dorsal; (**B**), same, lateral; (**C**), rostrum (setae omitted), lateral; (**D**), posterior half of pleon, lateral; (**E**), telson, dorsal; (**F**), same, posterior third, dorsal; (**G**), antennal scaphocerite, dorsal; (**H**), mandible, lateral; (**I**), same, mesial; (**J**), maxillule, lateral; (**K**), maxilla, lateral; (**L**), first maxilliped, lateral; (**M**), second maxilliped, lateral; (**N**), third maxilliped, lateral; (**O**), uropod (setae omitted), dorsal; (**P**), rostrum of holotype (setae omitted), lateral.

Hosts are known for *R. arabica* (soft corals (Nephtheidae), see below), *P. kallisto* (bushy black corals (Antipatharia), see Bruce [19]), *P. affinis* (several genera and species of feather stars (Crinoidea), see Holthuis [59] and Bruce [71]), and *A. brevicarpalis* (several genera and species of sea anemones (Actiniaria), see Lee & Koh [70], Anker & De Grave [74] and Ďuriš & Horká [10] and references therein), but not for *P. jugalis* and *P. canalinsulae* [72,73]. Thus, within the *Rubiyana–Ancylocaris* clade, the morphologically and ecologically most distinctive lineage is the monotypic genus *Ancylocaris*.

*Rubiyana* can be separated from *Periclimenes s. str.* and all other species presently referred to *Periclimenes* (except for *P. kallisto*), as well as *Ancylocaris* and all other satellite genera, by the above-mentioned, unique configuration of the uropod. The new genus also differs from *Periclimenes s. str.* as dactyli of the third to fifth pereiopods being simple (Figure 8F–K) vs. strongly biunguiculate in *Pericilimenes s. str.* [75–77]. Several other characteristics of *Rubiyana*, such as the elongate sixth pleonite and the absence of arthrobranch on the third maxilliped, may be revealed as important for differentiation between the new genus and numerous clades of *Periclimenes s. l.* Male specimens of *Rubiyana* are needed to verify the configuration of the appendix masculina of the second pleopod and the shape of the endopod of the first pleopod (simple or distally bilobed, as in *P. kallisto*).

***Rubiyana arabica* sp. nov.** (authored by Anker)
(urn:lsid:zoobank.org:act:96CA27B0-E5D7-4C3A-BC20-BC204EE1169A)
Figures 7, 8 and 9B,C

*Type material*: Saudi Arabia, Red Sea. Holotype: ovig. female (pocl 3.2 mm, cl 6.5 mm), RSRC (not deposited, see text), northwest of Farasan Islands, Red Sea Decade Expedition, Leg 2, Phase 2, sta. CHR0233, 17.387135–17.387619, 40.827209–40.828850, depth range 104–248.5 m, coll. Chimaera Dive Team/F. Benzoni and F. Barreca, 28.03.2022 (CHR0233-bycatch/OCX-073*). Paratype: ovig. female (pocl 2.9 mm, cl 6.0 mm), KAUST (not deposited, see text), about 15 km off Jeddah, Red Sea Decade Expedition, Leg 1, Phase 2, sta. CHR0198, 21.645553, 38.909941, depth 96 m, coll. Chimaera Dive Team/T.I. Terraneo, 23.02.2022 (CHR0198-Bio15A/OCX-198). *Designates sequenced specimen.

*Description*: Small-sized palaemonid shrimp (type material: pocl 2.9–3.2 mm) with cylindrical, not noticeably depressed or compressed body. Carapace smooth, glabrous. Rostrum (Figure 7A–C,P) moderately high, straight, horizontal, not reaching level of distal end of antennular peduncle, about as long as carapace (postorbital length); dorsal margin armed with eight to nine teeth, all of them situated anterior to posterior orbital margin; ventral margin straight, armed with two to three teeth on distal third; lateral carinae obsolete; supraorbital eaves absent. Orbit (Figure 7A,B) feebly developed; inferior orbital angle bluntly produced anteriorly. Antennal tooth (Figure 7B) well developed, submarginal, sharp, reaching level of distal end of inferior orbital angle. Hepatic tooth strong, sharp, situated slightly ventrally to antennal tooth (Figure 7A,B). Epigastric tooth (Figure 7A–C,P) distinctly removed from posterior-most tooth of rostral series, situated well posterior to level of posterior orbital margin and hepatic tooth, separated from carapace by suture. Supraorbital tooth absent. Ophthalmic somite without pronounced bec ocellaire (=interocular beak). Thoracic sternite 4 lacking median process. Epistome unarmed.

Pleon (Figure 7D) smooth, glabrous, with all pleura rounded ventrally; pleonite 3 with posterior margin slightly produced, overhanging dorsal surface of pleonite 4, forming low hump; pleonite 6 elongate, more than 2.5 times as long as high, about 1.2 times longer than telson (Figure 9C). Telson (Figure 7E) narrow, distinctly tapering from about 0.4 of length; dorsal surface with two pairs of spiniform setae, both located in posterior half and inserted at some distance from lateral margin; posterior margin triangular, with three pairs of spiniform setae, lateral ones very short, about 0.3 length of median ones and about 0.2 length of intermediate ones, latter much more robust.

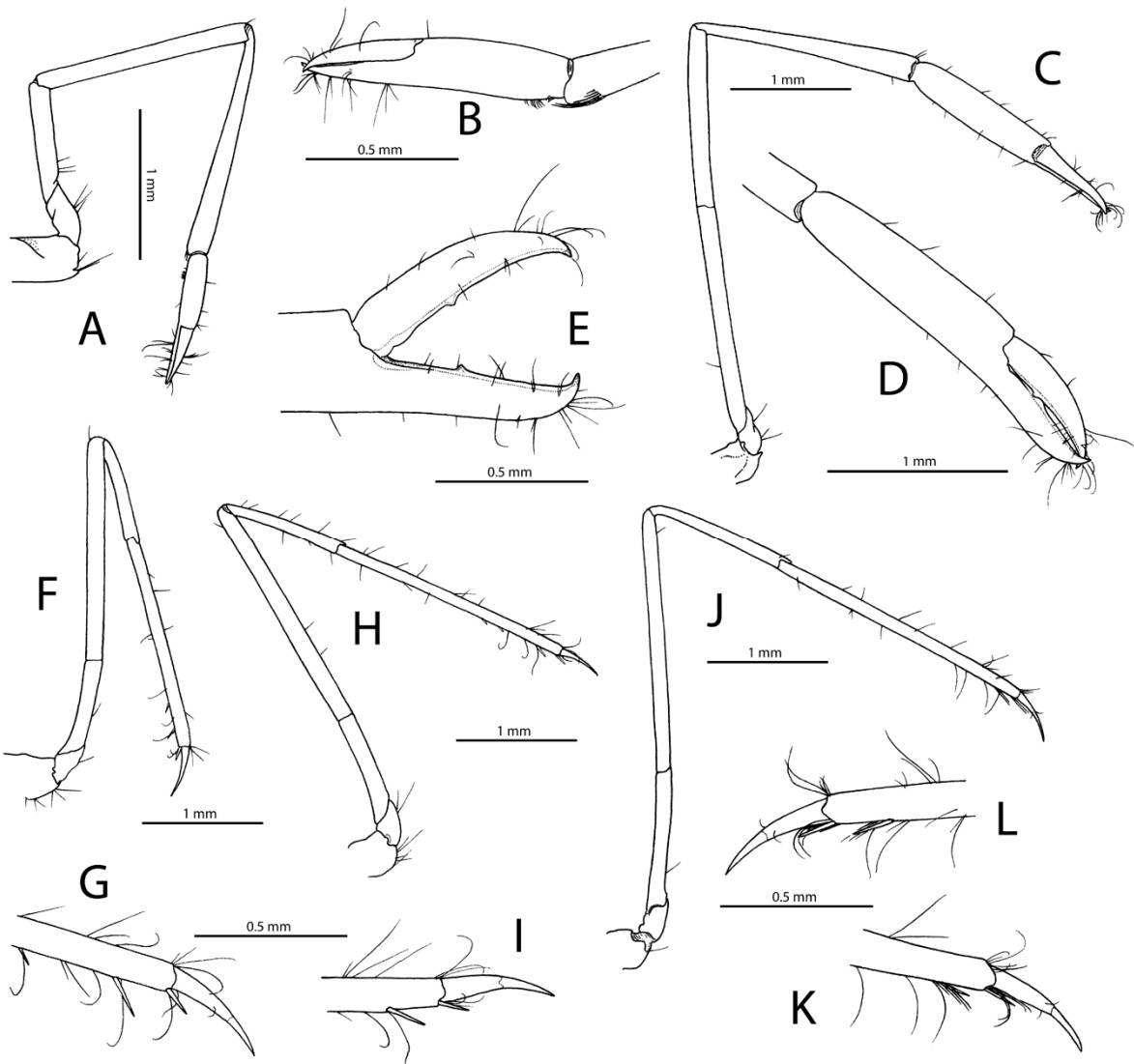

**Figure 8.** *Rubiyana arabica* gen. et sp. nov., paratype, ovigerous female (pocl 2.9 mm, cl 6.0 mm) from off Jeddah, Saudi Arabia, RSRC (not deposited, see text); (**A**), first pereiopod, lateral; (**B**), same, distal part of carpus and chela, mesial; (**C**), second pereiopod, lateral; (**D**), same, distal part of carpus and chela, lateral; (**E**), same, distal part of palm and fingers widely open, lateral; (**F**), third pereiopod, lateral; (**G**), same, distal half of propodus and dactylus, lateral; (**H**), fourth pereiopod, lateral; (**I**), same, distal half of propodus and dactylus, lateral; (**J**), fifth pereiopod, lateral; (**K**), same, distal half of propodus and dactylus, lateral; (**L**), same, distal half of propodus and dactylus, mesial.

Antennule (Figure 7A,B) with first article of peduncle robust, longer than length of second and third articles combined; lateral margin somewhat convex, terminating distally in strong acute tooth; distal margin strongly convex mesial to distolateral tooth; stylocerite small, acute, reaching 0.4 length of first article; ventromesial margin with small acute tooth; second and third articles subequal in length, much slenderer than first, unarmed; lateral flagellum biramous; accessory free ramus composed of two to three poorly defined subdivisions, with groups of aesthetascs. Antenna (Figure 7A,B,G) with basicerite short, distally armed with small sharp tooth; scaphocerite well developed; lateral margin straight, terminating in acute distolateral tooth, latter falling well short of bluntly angular distal margin of blade; carpocerite very short, not reaching 0.4 length of scaphocerite; flagellum normal, not thickened.

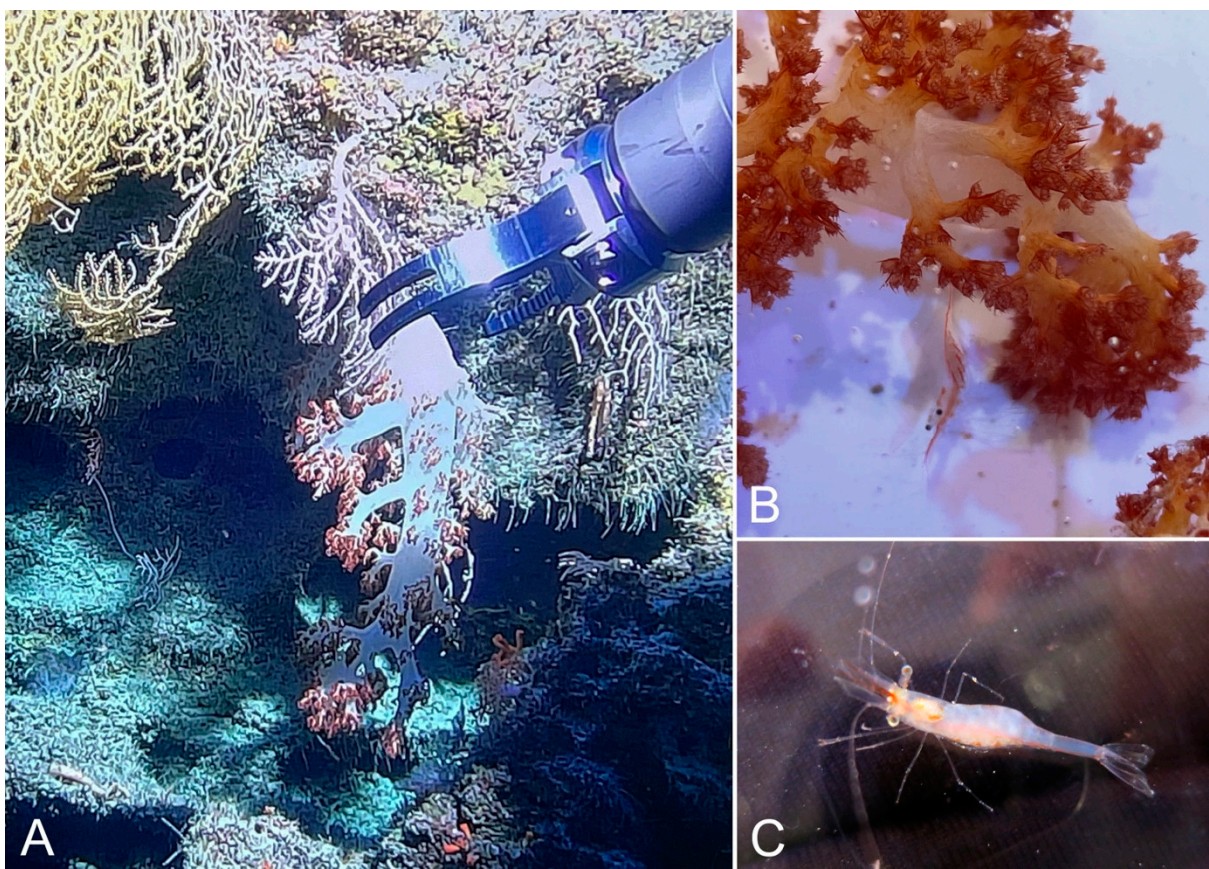

**Figure 9.** Nephtheid soft coral, *Dendronephthya* sp. (**A**), host of *Rubiyana arabica* gen. et sp. nov., paratype, ovigerous female (pocl 2.9 mm, cl 6.0 mm) from off Jeddah, Saudi Arabia, RSRC (not deposited, see text) during the collection of the host with a submersible manipulator at 96 m; (**B**), shrimp on its host after collection; (**C**), shrimp taken off the host (somewhat stressed), dorsolateral. Frame grab from a video recorded by OceanX team (**A**); photographs by T.I. Terraneo (**B**,**C**).

Mouthparts not especially modified, as illustrated (Figure 7H–N). Mandible (Figure 7H,I) without palp; molar process well-developed; incisor process with three teeth distally. Maxillule, maxilla and maxillipeds 1 and 2 (Figure 7J–M) without specific features, as described for genus. Maxilliped 3 (Figure 7N) with coxa bearing semicircular lateral plate; endopod moderately slender and setose; antepenultimate article not clearly separated from basis, without spiniform setae; penultimate article 0.6 times as long as antepenultimate article, much slenderer; ultimate article with stiff, non-spiniform setae; exopod short, not reaching 0.7 length of antepenultimate article; arthrobranch absent.

Pereiopod 1 (Figure 8A,B) slender, elongate; ischium about 0.6 length of merus; merus 0.9 length of carpus, somewhat angular ventrodistally; carpo-propodal grooming apparatus normally developed; chela about half-length of propodus; fingers about 0.7 length of propodus; dactylus conical, not subspatulate; cutting edges of both fingers unarmed. Pereiopods 2 subequal in length, similar in shape, slender, elongate (Figures 8C–E and 9C); coxa with stout ventromesial process; basis very short; ischium slender, almost 10 times as long as wide, 1.2 times as long as merus; merus about 0.8 length of carpus, unarmed; carpus subcylindrical, widening distally, about as long as chela; chela not inflated, elongate; palm cylindrical in cross-section, about 3.8 times as long as high, smooth, sparsely setose; fingers almost 0.6 length of palm, slightly gaping in distal half, with crossing fingertips; dactylus without lateral flange; cutting edge of each finger armed with small tooth at about 0.3 of finger length.

Ambulatory pereiopods (P3–5) similar, slender, elongate, somewhat increasing in length from third to fifth (Figure 8F–L); ischium ranging from about half-length of merus

(P3 and P4) to 0.6 length of merus (P5); merus ranging from 15.4 times (P3) to 18.4 times (P5) as long as wide, about twice as long as carpus; propodus as long as (P3 and P4) or slightly longer (P5) than merus; ventral margin of P3 and P4 with three to four spiniform setae on distal third or starting from 0.6 of propodal length, including distal one adjacent to dactylar base; ventromesial margin of P5 with two spiniform setae, including distal one adjacent to dactylar base; propodal cleaning brush reduced to two rows on distal ventrolateral surface; dactylus about 0.2–0.25 length of propodus, simple, moderately slender, with terminal unguis demarcated, slightly curved.

Male pleopods unknown. Uropod (Figure 7O) exceeding telson; protopod with two distally blunt lobes; exopod ovate-rectangular, rounded distally; distolateral tooth smooth; adjacent spiniform seta moderately strong; endopod narrowly ovoid, much smaller and shorter than exopod, its distal margin reaching to about 0.75 length of exopod.

*Colour pattern*: Largely translucent (somewhat milky opaque when stressed), with dissipated red chromatophores on carapace and pleon; conspicuous, continuous, red, longitudinal line present on sternum of carapace and pleon; antennular peduncles and pleopods with red bands; second pereiopods with yellow-red tinge distally; eggs pale yellow-green (Figure 9B,C).

*Etymology*: The specific epithet *arabica* (Latin adjective for Arabian/Arabic) refers to the Arabian region, which includes the Red Sea, where the new species occurs.

*Distribution*: Presently known only from the central and southern Red Sea, off the coast of Saudi Arabia.

*Ecology*: One of the specimens was found on a large soft coral of the genus *Dendronephthya* (Figure 9A,B), indicating a possible association between *R. arabica* and this group of cnidarians.

*Remarks*: As mentioned above, *R. arabica* is morphologically most similar to two species currently placed in *Periclimenes*, viz. *P. kallisto* and *P. affinis*. These two species were excluded from *Rubiyana* largely for phylogenetic reasons since their inclusion in the new genus would create a paraphyletic grouping (cf. Figure 2). On the other hand, applying the generic name *Ancylocaris* for the entire *Rubiyana–Ancylocaris* clade would result in a morphologically and ecologically wide, rather impractical genus. The inclusion of additional taxa with molecular data in the *Rubiyana–Ancylocaris* clade would certainly better resolve the relationships within this group. For the time being, both *P. kallisto* and *P. affinis* are retained in *Periclimenes* (see also above).

*Rubiyana arabica* can be separated from the antipatharian-associated *P. kallisto* by several morphological features, including the rostral formula with more numerous teeth on both margins, 9+1/2–3 (*vs.* 6–7+1/1 in *P. kallisto*); the epigastric tooth with distinct basal suture, i.e., semi-mobile (*vs.* fixed in *P. kallisto*); the much more elongate carpi of the first and second pereiopods; the second pereiopods subequal in size and with both chela fingers armed with a small tooth on their cutting edges (*vs.* somewhat unequal and with only the cutting edge of the pollex armed with small tooth in *P. kallisto*); and the ambulatory pereiopods terminating in simple, not subtly biunguiculate dactyli (*vs.* with subtly biunguiculate dactyli in *P. kallisto*) (Figures 7 and 8; cf. Bruce [19]: Figures 1–5). *Rubiyana arabica* is also distinguishable from the crinoid-associated *P. affinis*, for instance, by the carapace with a semi-mobile epigastric tooth distinctly removed from the posterior-most rostral tooth (*vs.* without epigastric tooth in *P. affinis*, in which all teeth on the mid-dorsal + rostral carina are equidistant); the ventral margin of the rostrum with two or three teeth (vs. with only one tooth in *P. affinis*); the much slenderer, subequal second pereiopods, with elongate, cylindrical carpus (*vs.* more robust, more unequal, and with cup-shaped carpus at least in the major cheliped, in *P. affinis*); the ambulatory pereiopods terminating in slender, gradually tapering dactyli (*vs.* with much stouter dactyli in *P. affinis*); and the telson with the dorsal spiniform setae distinctly removed from the margin (*vs.* submarginal in *P. affinis*) (Figures 7 and 8; cf. Holthuis [71] and Bruce [59]: Figure 2 and Figures 1–3, respectively). Finally, the new species can be easily separated from *P. jugalis* and *P. canalinsulae*, for example, from *P. jugalis* by the almost straight rostrum (*vs.* strongly arched in *P. jugalis*),

the cutting edge of the pollex with only one tooth (*vs*. two in *P. jugalis*), and the distinctly slenderer dactylus of the third pereiopod (Figures 7B and 8E,G; cf. Holthuis [72]: Figure 26); and from *P. canalinsulae* by the very different armature of the rostrum, the different position of the hepatic tooth relative to the antennal tooth, and the much shorter uropodal endopod (Figure 7B,O; cf. Bruce & Coombes [73]: Figure 3B,N).

### *Apopontonia* **Bruce, 1976**

### *Apopontonia falcirostris* **Bruce, 1976**

*Apopontonia falcirostris* Bruce [17]: 303, Figures 1–5; Bruce [78]: 3, 26; De Grave [79]: 121, Figure 1; Marin [80]: 218, Figure 1; Mitsuhashi & Chan [81]: 37 (key); Bruce [82]: 495.

*Material examined*: Saudi Arabia, Red Sea. One male (pocl 1.4 mm, cl 2.2 mm), KAUST, east of Farasan Islands, Red Sea Decade Expedition, Leg 2 Phase 2, sta. CHR0230, 16.931091, 41.140585, depth 88 m, inside large white sponge, coll. Chimaera Dive Team/F. Benzoni and F. Barreca, 25.03.2022 (CHR0230-Bio18B/OCX-114).

*Description*: See Bruce [17] for original description and De Grave [79] and Marin [80] for additional illustrations.

*Colour pattern*: Unknown.

*Distribution*: Indo-West Pacific: Red Sea, Madagascar, Maldives, Vietnam, Papua New Guinea, Australia [17,18,79,80,82], present study.

*Ecology*: The Red Sea specimen was found inside a non-identified, white, large sponge collected at 88 m, which is the deepest record for the species. The Madagascan type specimen was collected by a bottom trawl at 72 m and may have been also associated with a sponge (Bruce [17]: 311). All other specimens came from shallower water (i.e., scuba diving depths) and were associated with various sponges [18,79,82].

*Remarks*: The single specimen of *A. falcirostris* collected near the Farasan Islands represents the first record of this uncommon species and the monotypic genus *Apopontonia* [82] for the Red Sea and the Arabian region. The Red Sea specimen is a young male (with a short appendix masculina) that agrees rather well with the very detailed description of the female holotype of *A. falcirostris* provided by Bruce [17]. The few differences observed, e.g., in the proportions of the perfectly equal and symmetrical second pereiopods, are certainly due to the present specimen being male and of smaller size.

### *Bathymenes* **Kou, Li & Bruce, 2016**

*Bathymenes* Kou et al. [44]: 171; Ďuriš & Šobáňová [50]: 383.

*Diagnosis*: See Kou et al. [44] and Ďuriš & Šobáňová [50].

*Type species*: *Bathymenes alcocki* (Kemp 1922), originally described as *Periclimenes alcocki* Kemp, 1922, by original designation.

*Other species included*: See Kou et al. [44] and Ďuriš & Šobáňová [50] for full list of species included prior to the present study. Additional species included herein: *Bathymenes foresti* (Bruce, 1981) comb. nov.; *Bathymenes granuloides* (Hayashi, 1986) comb. nov.; *Bathymenes crosnieri* (Li & Bruce, 2006) comb. nov.; all three species originally described under *Periclimenes* (see discussion below).

*Hosts*: Sea pens (Pennatulacea); sea anemones (Actiniaria) carried by pagurids; sea urchins (Echinoidea); unknown for most species [44].

*Bathymetric range*: 187–824 m, but mainly in the depth range of 250–600 m.

*Distribution*: Indo-Pacific from East Africa to Chile; most species reported from the tropical western Pacific; not yet reported from the Red Sea.

*Remarks*: While comparing *Bruceclimenes* with *Bathymenes* (see above), some taxonomic incongruences were noted for the later genus. At least three species previously assigned to *Periclimenes* but with clear morphological connection to *Bathymenes* were not treated by previous authors [44,50]; for all three taxa, molecular data are currently lacking.

The species originally described as *Periclimenes crosnieri* by Li & Bruce [83] has all diagnostic characters of *Bathymenes*, as originally defined by Kou et al. [44], including the telson dorsally armed with four pairs of spiniform setae, the epigastric tooth clearly separated from the rostral series, and the second pereiopods finely granulated and with a lateral flange on the dactylus. Therefore, this species is herein transferred to *Bathymenes*, as *Bathymenes crosnieri* (Li & Bruce, 2006) comb. nov.

Along the same lines, *Periclimenes foresti* Bruce, 1981 and *Periclimenes granuloides* Hayashi, 1986 (perhaps a junior synonym of *P. foresti*, see Li et al. [48]) from the *P. foresti* complex (see above) display all diagnostic characters of *Bathymenes sensu* Ďuriš & Šobáňová [50], and are presumably closely related to *B. boucheti* (Li, Mitsuhashi & Chan, 2008) [18,47,48]. Therefore, they too are transferred to *Bathymenes*, as *Bathymenes foresti* (Bruce, 1981) comb. nov. and *B. granuloides* (Hayashi, 1986) comb. nov., respectively.

The generic diagnosis of *Bathymenes* in Ďuriš & Šobáňová [50], after inclusion of *B. boucheti*, and now also *B. foresti* and *B. granuloides*, is lacking a single autapomorphic feature immediately distinguishing it from other palaemonid genera. Most importantly, the dorsal armature of the telson in *Bathymenes*, as emended by Ďuriš & Šobáňová [50], now ranges from two to seven pairs of spiniform setae [44]. Variation also affects the dactylus of the ambulatory pereiopods, which can be simple or biunguiculate, the development of the cornea (usually reduced, but not always) and some other important features. The molecular analysis of two mitochondrial and two nuclear genes (Figure 2) provides no support for the distinction between the more typical species of *Bathymenes*, i.e., those with three or more pairs of dorsal spiniform setae on the telson, represented by *B. ngi* (Li, Mitsuhashi & Chan, 2008), *B. leptunguis* (Li, Mitsuhashi & Chan, 2008) and *B. sandybrucei* (Mitsuhashi & Chan, 2009), and some species formerly assigned to the *P. foresti* complex with two pairs of dorsal spiniform setae on the telson, such as *B. boucheti* (see also Horká et al. [8], and Ďuriš & Šobáňová [50]). However, more species of *Bathymenes* need to be included in future analyses to shed more light on the internal structure of the genus and perhaps propose a better classification scheme (two genera?), based on a combination of morphological and phylogenetic data.

**Supplementary Materials:** The following supporting information can be downloaded at: https://www.mdpi.com/article/10.3390/d15101028/s1, Table S1: GenBank deposition numbers for specimens sequenced in the present study + sequences downloaded from GenBank [8,21,28,29] used in the molecular analyses.

**Author Contributions:** ROV and sub-sampling and processing, S.V., F.B. (Francesca Benzoni), F.M., G.C., T.I.T. and F.B. (Federica Barreca); morphological analyses, A.A.; molecular and phylogenetic analyses, S.V.; taxonomic decisions and descriptions, A.A.; conceptualization and supervision, A.A. and F.B. (Francesca Benzoni); writing—original draft preparation, A.A. and F.B. (Francesca Benzoni); writing—review and editing, M.Q., A.A., S.V. and F.B. (Francesca Benzoni). F.B. (Francesca Benzoni), A.A., F.B. (Federica Barreca), S.V., T.I.T., F.M., M.R., C.M.D., V.P., A.A.E. and G.C. have read and commented on the manuscript. All authors have read and agreed to the published version of the manuscript.

**Funding:** The research expeditions onboard the R/V OceanXplorer were founded by Neom, NCW and OceanX. The authors are grateful to NCW for the invitation to participate in the RSDE and to OceanX ground and onboard staff and technical teams for their support. The material sampling and processing and KAUST team logistics and molecular analyses were supported by KAUST (award FCC/1/1973-50-01 and baseline research funds to F. Benzoni).

**Institutional Review Board Statement:** Not applicable.

**Data Availability Statement:** The genetic sequence data that support this study's finding are openly available in GenBank of NCBI under the accession numbers listed in Supplementary Material Table S1.

**Acknowledgments:** This research was undertaken in accordance with the policies and procedures of the King Abdullah University of Science and Technology (KAUST). Permission relevant for KAUST to undertake this research was obtained from the applicable governmental agencies in the Kingdom of Saudi Arabia. The authors acknowledge the Saudi Arabian authorities and the National Center for Wildlife in particular for supporting the OceanX expeditions onboard which we could sample for the present paper. For work in Neom in 2020, in addition to A.A.E., we are indebted to T. Habis, R. Khamis, P. Mackelworth, P. Marshall, J. Mynar and G. Palavacini for organising, coordinating, and facilitating the Deep Blue Expedition. For the Red Sea Decade Expedition, in addition to M.Q. and C.M.D., we thank National Center for Wildlife (NCW), J. E. Thompson, and N.C. Pluma Guerrero for organising, coordinating and facilitating the expedition. For the Red Sea Relationships Cultivation Expedition, the authors are grateful to M. Rodrigue and V. Pieribone for organising, coordinating and facilitating the expedition. The OceanX team, both in headquarters and onboard, is acknowledged for their operational and logistical support during the expedition, especially the captain and crew of OceanXplorer, and the Sub and ROV teams, to which we are deeply grateful for their patience, support and manipulator mastery. We thank the KAUST Sanger Sequencing Core Lab for helping with sequencing. Sammy De Grave (Oxford University Museum of Natural History) and Gustav Paulay (Florida Museum of Natural History) made stereomicroscopes equipped with camera lucida in laboratories under their supervision available for the preparation of line drawings.

**Conflicts of Interest:** The authors declare no conflict of interest.

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
