# Peer review of "Mesophotic and Bathyal Palaemonid Shrimp Diversity of the Red Sea, with the Establishment of Two New Genera and Two New Species†"

_diversity, doi:10.3390/d15101028_

Round 1

Reviewer 1 Report

The manuscript includes a precise description of two new genera with two new species of deep-sea palaemonid shrimps from the Red Sea. The presented drawings of the new species are of high quality. Taxonomic changes for three other deep-water species that do not occur in the Red Sea are also included.

The results are supported by both morphological and molecular data. A phylogenetic tree based on four molecular markers is presented.

There are only minor errors in the text. Some corrections are needed in Figure 2 and Table S1.
All my corrections and notes are in the attached pdf. file.

Author Response

Dear Reviewer 1,

Thank you for your thorough assessment of our manuscript. We have gone through your suggested changes and remarks and have modified the manuscript accordingly. We list hereafter your remarks and our responses.

R1: Line 46 and 48 “Error! Reference source not found.” For reference 4

Reply: reference 4 is present and properly formatted in the Reference list and available here https://www.semanticscholar.org/paper/Composition-of-the-deep-Red-Sea-macro-and-fauna.-T%C3%BCrkay/1e2e203d0e0847b839b9304647b7ba55b09bdc53  we are unsure of why the reference appears missing and suspect it must have been an error in the submission process

R1: line 65 “incorrect numbering of references”

Reply: we thank the R2 for pointing this out, however in the submitted Word version of the manuscript (and also in the one that can be downloaded from the Susy site) the reference numbers are 11, 12 and not 11, 11 as in the PDF generated at the time of submission

R1: Line 143 “correctly is a lower case letter "c"”

Reply: the text has been modified accordingly.

R1: Line 208 “a full stop is missing at the end of the sentence”

Reply: thank you, the full stop has been added

R1: Line 300 “Fig. 2 is correct ”

Reply: Indeed, thank you, text modified accordingly

R1: Line 302 “Fig. 3D is correct”

Reply: Indeed, thank you, text modified accordingly

R1: Line 315 “Anker & Corbari 2020 is correct ”

Reply: Indeed, thank you, text modified accordingly

R1: Line 368 “ Fig. 3A, B, E is correct”

Reply: Thank you, text modified accordingly

R1: Line 375 “Fig. 3D is correct”

Reply: Thank you, text modified accordingly

R1: Line 395 “Fig. 3E is correct”

Reply: Thank you, text modified accordingly

R1: Line 405 “foveolatus”

Reply: Thank you, text modified accordingly

R1: Line 419 “delete the bracket ”

Reply: Bracket deleted as suggested

R1: Line 425 “Michaelimenes, Okuno 2017 [20]”

Reply: 2017 has been added

R1: Line 528 “5A is correct”

Reply: Thank you, text modified accordingly

R1: Line 541 “6A, B ”

Reply: Thank you, text modified accordingly

R1: Line 547 “6A, B”

Reply: Thank you, text modified accordingly

R1: Line 697 “P. kallisto”

Reply: Text modified accordingly and genus abbreviation now corrected

R1: Lines 699-700 “Horká et al. [8]: figs. 2,3 ?”

Reply: The citation has now been added

R1: Line 712 “space”

Reply: space removed from the text

R1: Line 746 “9B, C”

Reply: Thank you, text modified accordingly

R1: Line 765 “Fig. 7A-C, P ???”

Reply: Thank you, text modified accordingly

R1: Line 789 “Fig. 7H, I is correct”

Reply: Thank you, text modified accordingly

R1: Line 801 “???”

Reply: Text now reads “(Figs. 8C–E, 9C);”

R1: Line 810 “Fig. 8F-L ?”

Reply: Text corrected as suggested

R1: Line 862, highlighted punctuation error

Reply: punctuation error now fixed

R1: Line 932 “ÄŽuriš & Šobáňová [50]”

Reply: taxonomic authority added

R1: Line 956 “A, and B in the map are not defined in the caption”

Reply: the reference to the panels in Figure 1 is now added

R1: Figure 2 “Miropandalus and Chlorotocella are representatives of Pandalidae. Specify it in the figure or in the caption.”

Reply: The two genera are actually currently placed in the Chlorotocellidae by Komai, T.; Chan, T.-Y.; De Grave, S. (2019), the text has been added in the figure legend “The Chlorotocellidae Komai, Chan & De Grave, 2019 family lineage is represented here by rep-resentative of the genera Miropandalus Bruce, 1983 and Chlorotocella Balss, 1914 at the top of the tree”

R1: Figure 2 the Reviewer reaquested modifying the figure to add “comb. nov.” and “sp. nov.”

Reply: We understand R1 request, however for once we have decided not to accept their valuable insights and correction as the Figure legend already clearly states the taxa names that are indeed “comb. nov.” and “sp. nov.”.

R1: Line 1033 “not in italics”

Reply: now not italicized, thank you

R1: Line 1040 “Table S1: Please separate (highlight) the families (Palaemonidae, Pandalidae, Stenopodidae) and arrange the species within them alphabetically. In accordance with the manuscript, list the references as numbers in square brackets. Correct these species name: Periclimenes wirtzi, Periclimenes boucheti, P. leptunguis, P. sandybrucei. These species are in other genera.”

Reply: Table S1 has been modified taking into account all suggested changes and additions with the notable exception of the family name Pandalidae actually replaced by the Chlorotocellidae name for the above mentioned reasons and in accordance with Komai, T.; Chan, T.-Y.; De Grave, S. (2019).

Reviewer 2 Report

This is a well written and excellently illustrated manuscript on new findings of palaemonid shrimps from the mesophotic and bathyal ecosystems in the Red Sea. Two new genera were established and two new species described. I agree with the taxonomic decisions made with regards to establishing the new taxa and the emendation of the diagnosis with regards to Michaelimenes.

Although it is a pity that some of the type material of the new species was lost, the authors provided enough information to warrant description of the species.

I made few corrections and added some comments in the text file attached.

Author Response

Dear Reviewer 2,

Thank you for your helpful assessment of our manuscript. We have gone through your suggested changes and remarks and have modified the manuscript accordingly. We list hereafter your remarks and our responses.

R2: line 65 “incorrect numbering of references”

Reply: we thank the R2 for pointing this out, however in the submitted Word version of the manuscript (and also in the one that can be downloaded from the Susy site) the reference numbers are 11, 12 and not 11, 11 as in the PDF generated at the time of submission

R2: Line 100 among the others – erease “the”

Reply: “and” erased

R2: Line 528 “Should be '5'.”

Reply: thank you, text modified accordingly

R2: Line 545 “It is much shorter than the chelae in the figures.”

Reply: Indeed, R2 is right, and the text has been modified to erase the incorrect statement altogether.

R2: Line 547 “6A, B”

Reply: Thank you, text modified accordingly

R2: Line 558 seta instead of setae

Reply: thank you, text modified accordingly

R2: Line 612 add “be”

Reply: text modified as suggested

R2: Line 623 “Holthuis (1951: p.34) mentiones the proximal excavation on the pollex in P. tenellus (also a deep water species): "The fixed finger is higher than the dactylus, the outer surface of the fingers is convex, the inner surface of each finger has a rather sharply defined elevated longitudinal

line. The space between these lines is strongly concave, so that the cutting edge lies in a rather deep hollow.”

Reply: We are grateful to R2 for raising this point. We modified the text and the new sentence now reads “This is especially true for P. tenellus, which possesses at least a shallow depression on the pollex of the second pereiopod chela (Holthuis 1951 [63]; Anker et al. 2014 [64]) and may well represent a western Atlantic species of Michaelimenes. In the absence of DNA data for these taxa, it is more prudent not to transfer P. tenellus and the remaining three species from Periclimenes to Michaelimenes. Nevertheless, they need to be considered in the comparison with the herein described new species of Michaelimenes.”

R2: Line 746 incorrect figure reference

Reply: Thank you, text modified to provide the correct figure reference

R2: Line 746 incorrect figure reference

Reply: Thank you, text modified to provide the correct figure reference

R2: Line 789 “Should be 'H'.” and “7H, I)”

Reply: Thank you, text modified accordingly

R2: Line 801 “Should be '9'.”

Reply: Text now reads “(Figs. 8C–E, 9C);”

R2: Line 810 “Should be 'L'.

Reply: Text corrected as suggested

R2: Line 862 “This is confusing”

Reply: text modified to clarify

R2: Line 979 “It is not visible in the figure with the resolution abailable.”

Reply: text modified in “attempt of collection of the sea anemone host, with the shrimp (male in B) silhouette in front of the retracted host in the middle panel (before suction took place)”

R2: Line 922 “palp” and “ventral”

Reply: the term has been changed to palp and the spelling error corrected.